# Homeostatic synaptic plasticity rescues neural coding reliability

Eyal Rozenfeld[1,2,4], Nadine Ehmann[3,4], Julia E. Manoim [1], Robert J. Kittel [3] ✉ & Moshe Parnas [1,2] ✉

To survive, animals must recognize reoccurring stimuli. This necessitates a reliable stimulus representation by the neural code. While synaptic transmission underlies the propagation of neural codes, it is unclear how synaptic plasticity can maintain coding reliability. By studying the olfactory system of *Drosophila melanogaster*, we aimed to obtain a deeper mechanistic understanding of how synaptic function shapes neural coding in the live, behaving animal. We show that the properties of the active zone (AZ), the presynaptic site of neurotransmitter release, are critical for generating a reliable neural code. Reducing neurotransmitter release probability of olfactory sensory neurons disrupts both neural coding and behavioral reliability. Strikingly, a target-specific homeostatic increase of AZ numbers rescues these defects within a day. These findings demonstrate an important role for synaptic plasticity in maintaining neural coding reliability and are of pathophysiological interest by uncovering an elegant mechanism through which the neural circuitry can counterbalance perturbations.

Animals encounter the same stimuli repeatedly and are able to recognize them over and over again. An essential requirement for the repeated identification of a stimulus is that its representation by the neural code will be reliably reproduced on each occasion. Two major approaches are used to describe neural coding[1]. The first is rate coding, where the firing rate of action potentials over a period of time is employed as the neural code. The second is temporal coding, where the precise timing and pattern of action potentials is utilized for coding. Often, a combination of the two is used as the coin of information for neural coding: the firing rate within a relatively small time window along with changes in firing rate over time.

Coding of an external stimulus by the nervous system begins at the sensory neurons. The code then undergoes transformations as it passes across synaptic contacts and propagates to higher brain regions. Chemical synaptic transmission is the major mode of fast information transfer between neurons. The response of a postsynaptic neuron depends on neurotransmitter release from synaptic vesicles (SVs) fusing with the plasma membrane at the presynaptic active zone

(AZ). SV fusion, in turn, is regulated by the complex interplay of various specialized AZ proteins, which determine neurotransmitter release sites and control release probability[2]. The molecular mechanisms of SV fusion are highly dynamic and can be modified on timescales ranging from milliseconds to days. Such presynaptic plasticity involves e.g. changes in $Ca^{2+}$ signals, SV availability, and activity-dependent modulations of the release machinery[3–6]. Importantly, synapses are also under homeostatic control. Homeostatic synaptic plasticity describes a phenomenon whereby pre- or postsynaptic adjustments maintain synaptic stability[7–10]. This evolutionarily conserved process rebalances destabilizing perturbations and preserves functionality in a changing environment. Correspondingly, dysfunctional homeostatic synaptic plasticity has been implicated in several neurological diseases[11]. Considering the many facets of synaptic plasticity, the question arises how neural coding reliability is affected by dynamic changes in neurotransmission[12–15].

The *Drosophila* olfactory system is highly amenable to genetic manipulations and is suitable to examine how synaptic function and

[1]Department of Physiology and Pharmacology, Sackler School of Medicine, Tel Aviv University, Tel Aviv 69978, Israel. [2]Sagol School of Neuroscience, Tel Aviv University, Tel Aviv 69978, Israel. [3]Department of Animal Physiology, Institute of Biology, Leipzig University, 04103 Leipzig, Germany. [4]These authors contributed equally: Eyal Rozenfeld, Nadine Ehmann. ✉e-mail: kittel@uni-leipzig.de; mparnas@tauex.tau.ac.il

plasticity affect neural code reliability. In flies, odors activate olfactory receptor neurons (ORNs). In general, each ORN expresses a single odorant receptor gene[16] (but note ref. [17]). Approximately 20–40 ORNs expressing the same receptor send their axons to a single glomerulus in the antennal lobe (AL)[18]. Approximately 2–5 second-order projection neurons (PNs), in turn, send their dendrites to a single glomerulus[19] and each PN samples all arriving ORNs[20]. The synaptic connection between an individual ORN and PN harbors ~10–30 release sites (scaling with glomerulus size) possessing a high average release probability (~0.75)[21]. The AL also contains multiglomerular inhibitory GABAergic and glutamatergic local neurons (iLNs) and excitatory cholinergic LNs (eLNs)[22–26]. The ORN-PN synapse is mainly modulated by the GABAergic iLNs[24,27]. Thus, the *Drosophila* ORN-PN connection is a system with a reliable synapse, which can be genetically modified and examined in the context of physiological odor stimuli.

By performing in vivo whole-cell patch-clamp recordings from over 1200 neurons, we show that knocking down the AZ $Ca^{2+}$ channel Cacophony (Cac) specifically in ORNs decreases neural coding reliability in the postsynaptic PNs to physiological odor stimuli. Our results demonstrate that interfering with SV release probability from ORN AZs affects the PN neural code in three ways. First, the initial transient phase of the odor response, when PNs reach their highest firing rates, becomes more variable due to reduced recruitment of iLNs. Second, the onset of the odor response is delayed and more variable. Third, the temporal dynamics of the odor response become less reliable; this does not involve any circuit motifs and results from monosynaptic effects. In line with its ethological relevance, reducing the reliability of neural coding leads to a reduction in the behavioral ability to correctly classify an olfactory stimulus. Surprisingly, however, we find that decreased expression of Cac at ORN AZs affects neural code reliability only at high stimulus intensities. This differential effect is due to homeostatic synaptic plasticity, which sets in within a day. Thereby, the formation of additional AZs compensates the reduced neurotransmitter release probability and rescues coding reliability of PNs. Thus, our work uncovers a new role for homeostatic synaptic plasticity in maintaining stable neural coding and reliable behavior.

## Results

### ORN SV release probability affects olfactory coding reliability at high odor intensities

In order to investigate how molecular properties of the AZ influence the olfactory neural code, we recorded postsynaptic PN responses to odor presentation with in vivo whole-cell current-clamp recordings (Fig. 1A). We used RNA interference (RNAi) in ORNs to knock down Cac, the pore-forming subunit of the *Drosophila* $Ca_v2$-type $Ca^{2+}$ channel, which mediates $Ca^{2+}$-influx at the AZ[28,29]. Since different odors elicit different firing rates and temporal dynamics in ORNs[30] we tested PN responses to five different odors (Isopenthyl acetate, linalool, 2-heptanone, ethyl acetate, and isobutyl acetate) and used high odor concentrations to minimize variations in ORN responses. To examine stimulus-evoked response reliability we recorded PN responses to 10 applications of each odor (Figs. 1A–C and S1). *Orco-GAL4* drove *UAS-cac^{RNAi}* specifically in ORNs and *GH146-QF*, which labels ~60% of PNs[31], was used to drive *QUAS-GFP* for targeted patch-clamp recordings. We confirmed RNAi efficiency for *cac* by performing in vivo $Ca^{2+}$ imaging via two-photon microscopy during odor stimulation. *Orco-GAL4* was used to drive *UAS-cac^{RNAi}* along with *UAS-GCaMP6f*. As expected, $Ca^{2+}$ signals were significantly reduced in ORN terminals when *cac* was targeted by RNAi (Fig. S2A). In addition, direct staining against Cac demonstrated reduced protein levels (Fig. S2B). *cac^{RNAi}* significantly reduced odor-induced PN activity (Figs. 1B, C and S2C), illustrating that SV release from ORNs was decreased, though not completely abolished.

A number of different methods can be used to examine neural code reliability. Correlation-based measurement judges the reliability of the temporal dynamics of neural activity by analyzing the activity

time-series as a single entity[32] (Fig. S3A, C). Thus, this analysis examines the response pattern over time, but does not consider the response magnitude. Therefore, correlation measurements do not account for variations in firing rate at a specific time point (Fig. S3A, C). For correlation analysis, we calculated correlations between the 10 repetitions of each PN-odor pair with a given bin size (from 1 to 200 ms, Fig. S3A, C). Another measure of temporal code reliability is the latency of the first spike following a given stimulus. First spike latency has been found to encode sensory information across different sensory modalities and organisms[33–35] and, in particular, odor identity and intensity in *Drosophila*[36]. The standard deviation of response magnitude examines response reliability for a given time point or a given firing rate and does not account for the temporal dynamics (Fig. S3B, C)[37]. For this analysis, we calculated the number of spikes and variability for each individual bin and pooled the mean variance value for all bins having the same frequency across neurons, odors, and time (Fig. S3B, C).

At high odor concentration, correlation analysis using spike trains that were integrated at increasingly wider temporal windows (Fig. S3A), revealed that for wild type (*wt*) flies the response reliability (correlation value) became saturated at ~20 ms (Fig. 1D, E, see "Methods" section), consistent with previous reports[38]. Here, the integration window required to reach saturation was defined using the Youden's index[39]. Knockdown of *cac* decreased overall correlation values and increased the temporal integration window (Figs. 1D, E and S4A, B). A similar result was obtained when the Euclidean distance was used to measure similarity (Fig. S5). This demonstrates that molecular perturbations of ORN AZs decrease the temporal reliability of PN responses. Standard deviations of PN firing rate, as a function of the mean firing rate during a 20 ms time window, also revealed a decrease in PN coding reliability in *cac^{RNAi}* flies (Figs. 1F and S4C) with PNs showing increased variability especially at high firing rates (Fig. 1F).

PN somata are distant from the action potential initiation site[40]. As a result, action potentials recorded at the soma in current-clamp mode are small, often reaching only 5 mV in amplitude[40]. Such small action potentials are frequently not identified during the rising phase of the odor response. To better detect the first action potential, indicating the response onset, we repeated this set of experiments in voltage-clamp mode. Indeed, voltage-clamp recordings enabled detection of more action potentials in the first 25 ms of the odor response and the latency of the first action potential was shorter (Fig. 1G–I). As with the other neural coding reliability measurements, this analysis revealed that knocking down *cac* resulted in lower reliability, reflected by both increased latency and increased standard deviation of the first action potential (Figs. 1J, K and S6).

High odor concentrations elicit high firing rates in ORNs that may reach up to 350 Hz[41]. To examine if the effect of *cac^{RNAi}* persists when ORNs fire at lower rates, we repeated the above experiments using a weaker odor intensity. Interestingly, we found that under these conditions, response magnitude and coding reliability by PNs were not decreased by reduced Cac expression (Fig. 2A–D). Taken together, our data show that reducing SV release from ORNs affects the reliability of the postsynaptic odor response. However, this effect is only evident during the presentation of a strong olfactory stimulus.

The above results suggest that high-frequency synaptic transmission is most sensitive to manipulations of ORN AZs, consistent with previous reports[42]. High firing rates usually occur at the beginning of the odor response within the first 200 ms (Fig. 1B, C). Thus, we examined the effects of *cac^{RNAi}* during the presentation of a strong olfactory stimulus without the initial phase of the odor response. The initial phase is characterized by a strong increase in firing rate that is highly correlated. Thus, as expected, removing the first 200 ms of the odor response from the analysis resulted in an overall decrease in correlation (Fig. 2E compare to Fig. 1D), which was further reduced by *cac^{RNAi}* (Fig. 2E, F). In contrast, the increased standard deviation of PN

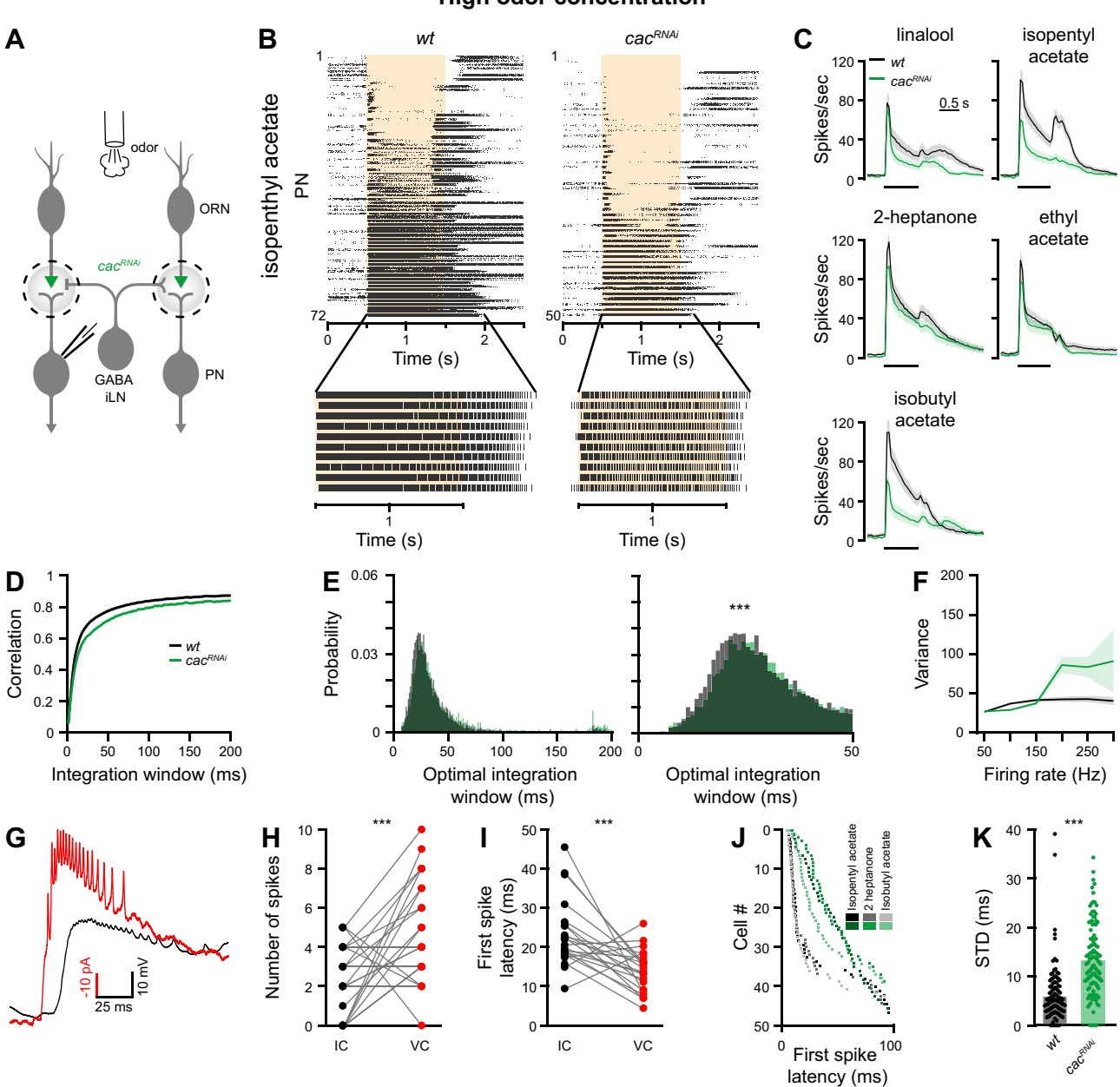

**Fig. 1 | Reducing SV release probability reduces coding reliability at high stimulus intensities. A** *cac*^RNAi was expressed in ORNs and recordings were made from PNs. PN responses were measured for high odor concentration in 2–4-day-old flies. **B** Top, Raster plot of PN population odor responses to isopentyl acetate in *wt* flies (left, *n* = 72 flies) and *cac*^RNAi in ORNs (right, *n* = 50 flies). Bottom, the 10 repetitions of a single PN are presented. The shaded area indicates the odor stimulus. **C** Peristimulus time histogram (PSTH) of PN population responses to five odors (shaded areas represent SEM, odor pulse is labeled with a black bar). Knockdown of *cac* (green) in ORNs resulted in decreased PN odor responses. (*n* = 48-72 flies). **D** Temporal reliability analysis. Pairwise correlations for each odor-neuron combination were pooled across all odors for the data presented in Fig. S1. *cac*^RNAi in ORNs reduces correlation values. **E** Left, the curve saturation point was calculated for each odor-neuron combination and pooled across all odors for the data presented in Fig. S1. Right, Left curves are presented at a smaller scale. *cac*^RNAi in ORNs shifts the optimal temporal integration window of PNs as evident by a larger integration window spread (left) and a peak shift of -10 ms (right). Two-sided permutation

test, *p* < 0.001. **F** Firing-rate reliability analysis for the data presented in Fig. S1. Increased rate variability for *cac*^RNAi flies is observed for high firing rates. Error bands represent SEM. **G** Example recordings performed in current clamp (IC, black) and in voltage clamp (VC, red). VC trace is inverted for presentation purposes. **H** PN firing-rate during the first 25 ms of the response to isopentyl acetate in current clamp (IC, black) vs. voltage clamp (VC, red). Each dot represents an individual fly (*n* = 22). The VC configuration enabled detection of more spikes. Two-sided two sample paired *t*-test, *p* = 0.001. **I** First spike latency of PNs in response to isopentyl acetate in IC (black) vs. VC (red). Dots represent individual flies (*n* = 22). Two-sided two sample paired *t*-test, *p* < 0.001. **J** First spike latency of PNs in response to the indicated odors for *wt* and *cac*^RNAi flies (*n* = 50). Each dot represents the mean first spike latency for 10 trials of a given neuron. Data were obtained in VC and PNs that did not spike within 100 ms after stimulus onset were omitted. **K** First spike jitter (standard deviation, STD) of PN odor responses, pooled across all odors, for *wt* and *cac*^RNAi flies (*n* = 50). Two-sided permutation test, *p* < 0.001. For all panels ***p* < 0.001, for detailed statistical analysis, see Table S1.

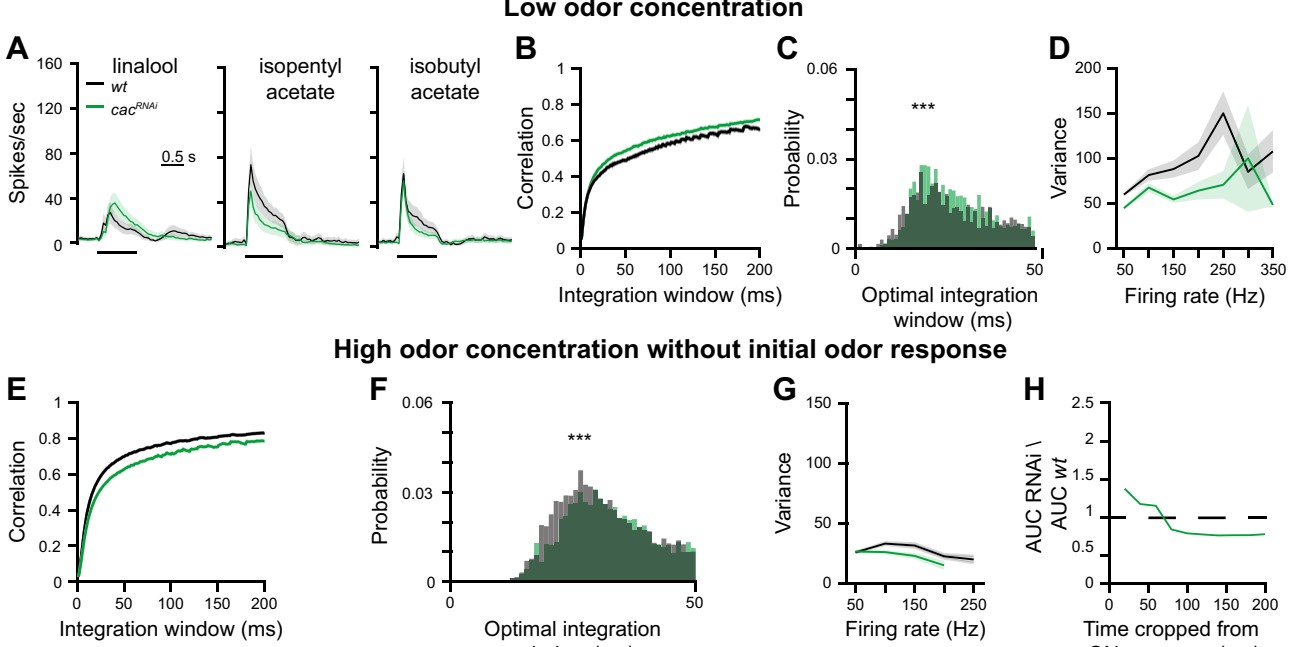

**Fig. 2 | Reducing SV release probability does not affect coding reliability at low stimulus intensities. A** PSTH of PN population responses to three low concentration odors examined as indicated (shaded areas represent SEM, odor pulse is labeled with a black bar). Spike trains were binned using a 50 ms time bin. Knockdown of *cac* (green) in ORNs did not affect PN odor responses. A final odor dilution of $5 \times 10^{-4}$ was used (*n* = 50 flies). *Orco-GAL4* drove the RNAi construct and *GH146-QF* drove *QUAS-GFP*. **B, C** Temporal reliability analysis for data in panel A. *cac^RNAi^* in ORNs did not reduce correlation values. Two-sided permutation test, *p* < 0.001. **D** Firing-rate reliability analysis for data in panel A. *cac^RNAi^* in ORNs did not increase firing rate variability. Error bands represent SEM. **E, F** Temporal reliability analysis for data in Fig. S1. Removing the first 200 ms of the odor response leads to

an overall decrease in correlation. A shift of the peak of the curve is observed for the sustained odor response in *cac^RNAi^* flies. Two-sided permutation test, *p* < 0.001. **G** Firing-rate reliability analysis for data in Fig. S1 without the first 200 ms of the odor response. Contrary to the increased variation observed for the entire odor response (Fig. 1F), *cac^RNAi^* in ORNs did not increase the variability for the sustained odor response although high firing-rates were still obtained. Error bands represent SEM. **H** The ratio between the area under the curve (AUC) of the rate code variability in *cac^RNAi^* relative to *wt*. The initial response was progressively cropped from the analysis. The difference in rate code variation of *wt* and *cac^RNAi^* flies gradually decreases over the first 200 ms of the odor response. For all panels ****p* < 0.001, for detailed statistical analysis, see Table S1.

firing rates that was observed upon *cac^RNAi^* was now completely abolished. No difference in the standard deviation was observed between *wt* and RNAi flies even though high firing rates were still obtained in the sustained phase of the odor response (Fig. 2G). The effect of *cac^RNAi^* was not equally distributed throughout the initial phase of the odor response; rather it was most notable when PNs reached their peak firing rate, ~20 ms after the response onset (Fig. 2H). To verify that the observed effect of *cac^RNAi^* indeed arises from an impairment of SV release, we also tested Synapsin (Syn), a SV-associated phospho-protein involved in the organization of SV pools[43–45]. Expression of *UAS-syn^RNAi^* in ORNs resulted in a similar reduction of coding reliability by PNs (Figs. S7 and S8). Previous work showed that repeated odor exposure increases the trial-to-trial correlations of odor responses[46]. It was therefore of interest to examine whether *cac^RNAi^* would also lead to a lower reliability under these conditions. We too observed an increase in the trial-to-trial correlations which was lost upon *cac^RNAi^* in ORNs (Fig. S9).

Taken together, the above results suggest that two different mechanisms underlie the reduced response reliability of PNs, one affecting the reliability of the temporal firing rate dynamics and the other affecting the reliability of the absolute firing rate magnitude.

### Knockdown of *cac* reduces release probability and increases synaptic latency and jitter

Reducing Cac expression in ORNs significantly reduced PN spiking (Fig. 1). Nevertheless, PNs maintained a substantial response to odors, demonstrating that synaptic transmission from ORNs to PNs remained principally functional. Cac is a key regulator of SV release probability at

*Drosophila* neuromuscular AZs. Its functional disruption in a temperature-sensitive mutant, RNAi-mediated knockdown, and impaired AZ channel clustering all lead to greatly decreased evoked excitatory postsynaptic current (eEPSC) amplitudes at the neuromuscular junction (NMJ), especially at low stimulation frequencies[28,29,47]. We were therefore intrigued by the observation that *cac^RNAi^* affected PN coding reliability only at high and not at low ORN firing rates. To understand this phenomenon, we characterized functional properties of the ORN-PN synapse in *wt* and *cac^RNAi^* animals (Fig. 3A). Surprisingly, the average eEPSC amplitude of *cac^RNAi^* did not differ from *wt* (Fig. 3B). However, *cac* knockdown led to increased paired-pulse facilitation at short inter-pulse intervals (Fig. 3C, D), consistent with a decrease in SV release probability upon reduced expression of AZ Ca$^{2+}$ channels[47]. In line with a drop in release probability, *cac^RNAi^* synapses also displayed significantly elevated synaptic delay (Fig. 3E), which was accompanied by an increase in the jitter of eEPSC (Fig. 3F). Interestingly, both parameters progressively increased with stimulation frequency in the transgenic animals (Fig. 3E, F). Within our sampling range (1, 10, 20, and 60 Hz), the effect of presynaptic *cac^RNAi^* on these temporal parameters was most pronounced at 60 Hz.

Taken together, these results suggest a homeostatic mechanism at the ORN-PN synapse, which compensates for the drop in release probability caused by presynaptic *cac^RNAi^* to maintain normal eEPSC amplitudes. However, this compensation does not cover the temporal properties of synaptic transmission. The reduced release probability of *cac^RNAi^* ORNs generates longer synaptic latencies with larger eEPSC jitter, consistent with increased first spike latencies and jitter during odor application. Thus, while the ORN-PN synapse is surprisingly

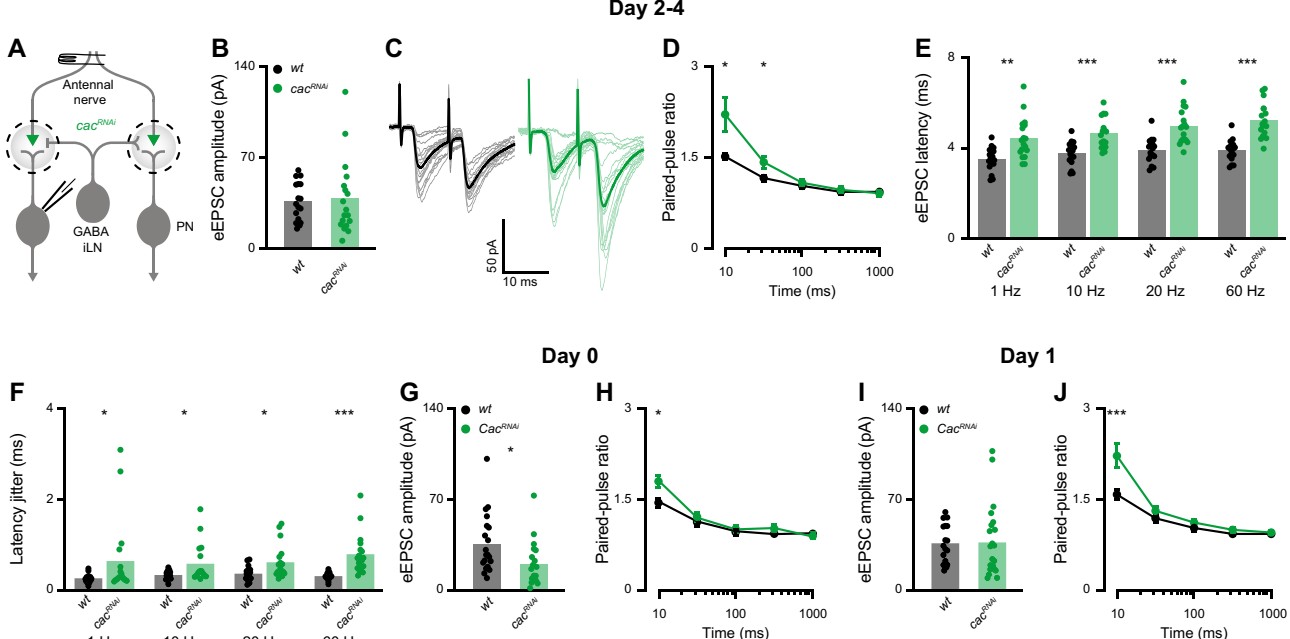

**Fig. 3 | Presynaptic *cac* knockdown induces homeostatic plasticity.**
**A** Experimental scheme. *UAS-cac$^{RNAi}$* was expressed in ORNs using *Orco-GAL4* and whole-cell patch clamp recordings were made from PNs labeled by the *GH146-QF* driver line. The antennal nerve was stimulated with a suction electrode. **B** The average eEPSC amplitude was unaltered by *cac$^{RNAi}$* for 2–4-day-old flies (1 Hz stimulation frequency; *wt* n = 17 and *cac$^{RNAi}$* n = 18 flies). Two-sided permutation test, p = 0.789. **C** Examples traces of eEPSC evoked by paired-pulse stimulation with an inter-pulse interval of 10 ms. **D** Quantification of the paired-pulse ratio at different inter-stimulus intervals for 2–4-day-old flies (10 ms, 30 ms, 100 ms, 300 ms, and 1000 ms). *cac$^{RNAi}$* significantly increased paired-pulse facilitation at short inter-pulse intervals [*wt*, n = 20 (10 ms), n = 21 (30 ms), n = 22 (100, 300, 1000 ms); *cac$^{RNAi}$*, n = 25 (10 ms), n = 24 (30 ms), n = 28 flies (100, 300, 1000 ms)]. Error bars represent SEM. Two-sided permutation test, p = 0.01 (10 ms), p = 0.034 (30 ms). **E, F** Average eEPSC latencies (**E**) and average eEPSC jitter (**F**) at 1, 10, 20, and 60 Hz stimulation for *wt* (n = 17) and *cac$^{RNAi}$* (n = 18) flies. A significant increase was observed for all tested stimulation frequencies. Two-sided permutation test and two-sided two

sample *t*-test, p = 0.001 (1 Hz latency), p < 0.001 (10 Hz latency), p < 0.001 (20 Hz latency), p < 0.001 (60 Hz latency and jitter), p = 0.044 (1 Hz jitter), p = 0.016 (10 Hz jitter), p = 0.01 (20 Hz jitter). **G** The average eEPSC amplitude was reduced by *cac$^{RNAi}$* for 0-day-old flies (1 Hz stimulation frequency; *wt* n = 20 and *cac$^{RNAi}$* n = 19 flies). Two-sided permutation test, p = 0.016. **H** Quantification of the paired-pulse ratio at different inter-stimulus intervals for 0-day-old flies (10 ms, 30 ms, 100 ms, 300 ms, and 1000 ms). *cac$^{RNAi}$* significantly increased paired-pulse facilitation at short inter-pulse interval [*wt*, n = 20 (10 ms), n = 21 (30, 100, 300, 1000 ms); *cac$^{RNAi}$*, n = 19 flies]. Error bars represent SEM. Two-sided permutation test, p = 0.025. **I** The average eEPSC amplitude was unaltered by *cac$^{RNAi}$* for 1-day-old flies (1 Hz stimulation frequency; *wt* n = 17 and *cac$^{RNAi}$* n = 18 flies). **J** Quantification of the paired-pulse ratio at different inter-stimulus intervals for 1-day-old flies (10 ms, 30 ms, 100 ms, 300 ms, and 1000 ms). *cac$^{RNAi}$* significantly increased paired-pulse facilitation at short inter-pulse interval [*wt*, n = 20 (10 ms), n = 21 (30 ms), n = 22 (100, 300, 1000 ms); *cac$^{RNAi}$*, n = 22 flies]. Error bars represent SEM. Two-sided permutation test, p < 0.001. For all panels *p < 0.05, **p < 0.01, ***p < 0.001, for detailed statistical analysis, see Table S1.

resilient to Ca$^{2+}$ channel perturbations, *cac$^{RNAi}$* delays synaptic transmission and reduces the temporal precision of synaptic signaling.

## A homeostatic increase in AZ numbers compensates for the drop in release probability

So far, all experiments were carried out on 2–4-day-old flies. To obtain more information on the homeostatic regulation of the ORN-PN connection, we next addressed the time course of the synaptic adjustment. At day 0 (<24 h post pupal eclosion), *cac$^{RNAi}$* caused a reduction in both release probability and eEPSC amplitudes (Fig. 3G, H). Moreover, *cac$^{RNAi}$* synapses responded in only 19 out of 40 cases irrespective of stimulus intensity (vs. 20 out of 20 in *wt*). In contrast, at day 1 (24–48 h post pupal eclosion), eEPSC amplitudes were already elevated to *wt* levels (Fig. 3I, J) and the transmission success rate reached 100%. When presented with a high odor concentration at day 0, *cac$^{RNAi}$* flies displayed decreased firing rates and reduced coding reliability compared to *wt* (Fig. 4A–E) and to 2–4-day-old flies (Fig. 1C–F). However, when a low odor concentration was applied, both firing rates and neural code reliability remained severely impaired (Fig. 4F, G), contrary to the compensation observed in 2–4-day-old flies (Fig. 2A–D). Thus, homeostatic synaptic plasticity took place within the first day post eclosion to rescue neural activity and coding reliability at low stimulus intensity but failed to do so at high odor concentrations.

Next, we turned to the mechanism underlying the homeostatic synaptic change responsible for maintaining normal eEPSC amplitudes. In principle, the drop in release probability caused by *cac$^{RNAi}$* could be counterbalanced by an increase in the number of presynaptic release sites or an increase in quantal size. Quantal size describes the postsynaptic response to the fusion of an individual SV with the AZ membrane. This parameter is reflected by the amplitude of spontaneously occurring miniature excitatory postsynaptic currents (minis) and can be influenced by the number and identity of postsynaptic receptors. We recorded similar mini frequencies and amplitudes at *wt* and *cac$^{RNAi}$* synapses (Fig. 5A, B) demonstrating an unaltered quantal size and, in turn, suggesting a homeostatic addition of release sites. In line with a corresponding increase in the number of postsynaptic receptor fields, puffing nicotine onto PNs elicited larger currents (Fig. 5C).

Based on these results, we sought to match the functional increase of release sites in *cac$^{RNAi}$* to a structural correlate. To this end, we performed immunostainings of the antennal lobe and imaged Bruchpilot (Brp), a core component of the AZ cytomatrix (CAZ)[47], via confocal microscopy. To restrict our analysis to ORN-PN synapses, we co-labeled postsynaptic PNs via GFP and quantified Brp in the overlapping regions (Fig. 5E). Consistent with the electrophysiological estimate of release site addition, the number of Brp punctae was significantly increased upon *cac* knockdown (Fig. 5D). Moreover, while

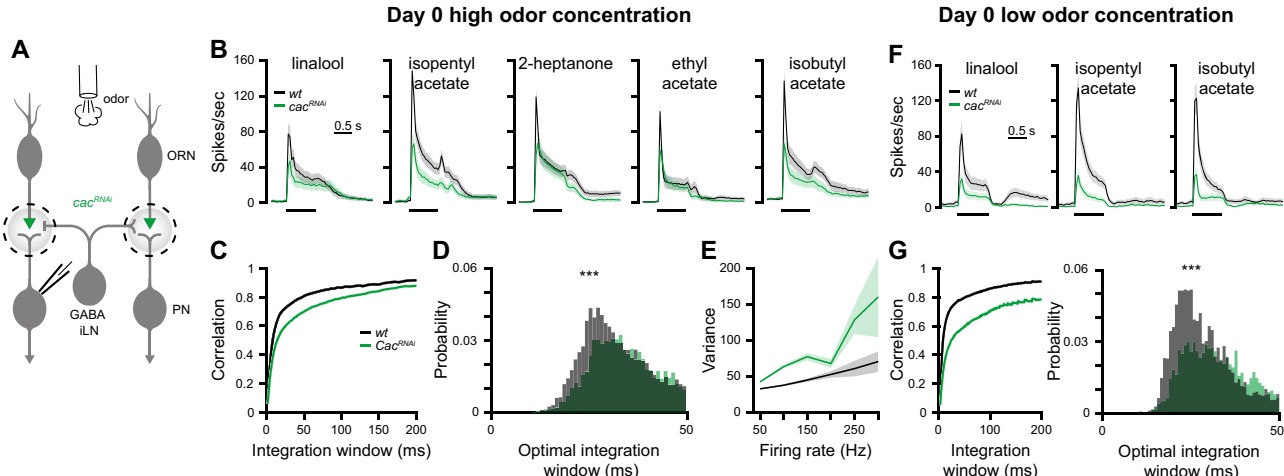

**Fig. 4 | Neural coding reliability is not rescued by homeostatic plasticity in newly eclosed flies. A** Experimental scheme. *UAS-cac^RNAi* was expressed in ORNs using *Orco-GAL4* and whole-cell patch clamp recordings were made from PNs labeled by the *GH146-QF* driver line. PN responses were measured for both high and low odor concentrations in 0-day-old flies. **B** PSTH of PN population responses to five high concentration odors examined as indicated (shaded areas represent SEM, odor pulse is labeled with a black bar) in 0-day-old flies. Spike trains were binned using a 50 ms time bin. Knockdown of *cac* (green) in ORNs resulted in decreased PN odor responses. A final odor dilution of $5 \times 10^{-2}$ was used (*n* = 50–57 flies). *Orco-GAL4* drove the RNAi construct and *GH146-QF* drove *QUAS-GFP*. **C**, **D**. Temporal reliability analysis (as in Fig. 1D, E) for data in panel B. *cac^RNAi* in ORNs reduced

correlation values. Two-sided permutation test, *p* < 0.001. **E** Firing-rate reliability analysis (as in Fig. 1F) for data in B. *cac^RNAi* in ORNs increased firing rate variability. Error bands represent SEM. **F** PSTH of PN population responses to three low concentration odors examined as indicated (shaded areas represent SEM, odor pulse is labeled with a black bar) in 0-day-old flies. Spike trains were binned using a 50 ms time bin. Knockdown of *cac* (green) in ORNs resulted in decreased PN odor responses. A final odor dilution of $5 \times 10^{-4}$ was used (*n* = 49–50 flies). *Orco-GAL4* drove the RNAi construct and *GH146-QF* drove *QUAS-GFP*. **G** Temporal reliability analysis (as in Fig. 1D, E) for data in panel F. *cac^RNAi* in ORNs reduced correlation values. Two-sided permutation test, *p* < 0.001. For all panels ***p* < 0.001, for detailed statistical analysis, see Table S1.

the average cluster size remained unchanged in *cac^RNAi*, the signal intensity of individual punctae was significantly elevated (Fig. 5D), reflecting an increased local Brp concentration and possibly the addition of CAZ units below the diffraction limit[48]. Thus, a homeostatic increase in the number of Brp-positive AZs counterbalanced the decreased transmitter release probability caused by reduced Cac expression. As expected from knocking down *cac*, this homeostatic plasticity did not appear to be accompanied by increased Ca^2+ influx. Using two-photon Ca^2+ imaging, we recorded similar responses in recently eclosed (day 0; Fig. S10) and 2–4-day-old *cac^RNAi* flies (Fig. S2A).

**Monosynaptic effects and circuit activity underlie distinct features of coding reliability**
Next, we examined whether the diminished coding reliability caused by *cac^RNAi* is due to monosynaptic mechanisms or instead arises from circuit effects. GABAergic iLNs also receive their input from ORNs and inhibit these presynaptically. Whereas PNs are uniglomerular and receive homogenous ORN input, iLNs are multiglomerular and receive heterogeneous ORN input. PNs respond to ORN activity in a non-linear manner[21,37,49]. In contrast, GABAergic iLNs respond linearly to ORN activity[49], are more sensitive to reduced input from ORNs[26], and show a stronger transiency in their odor response[50]. Thus, we hypothesized that at least some of the observed reduction in PN reliability following presynaptic manipulation might arise from reduced recruitment of iLNs. In particular, this may explain the increased standard deviation in *cac^RNAi* that no longer occurred in the absence of the initial phase of PN odor responses (Fig. 2G, H). To examine how *cac* knockdown affects iLN activity, we compared *wt* and presynaptic *cac^RNAi* flies (Fig. 6A). *Orco-GAL4* again drove *UAS-cac^RNAi* and *449-QF* was used to express *QUAS-GFP* in iLNs[51] (Fig. 6A). Reducing Cac protein levels in ORNs resulted in an almost complete abolishment of iLN odor responses in both 0 and 2–4-day-old flies (Fig. 6B, C). Similar to PNs, *cac^RNAi* reduced the SV release probability at the ORN-iLN synapse (Fig. 6D). However, in contrast to ORN-PN transmission, we observed no homeostatic

rescue of iLN responses to odor stimulation even in 2–4-day-old flies. Correspondingly, confocal imaging of the ORN-iLN connection revealed no increase in the number of Brp punctae upon *cac^RNAi* (Fig. S11).

These results demonstrate that knocking down *cac* in ORNs severely impairs the activation of iLNs. We therefore examined the effects of blocking GABAergic transmission on PN reliability. To this end, we exposed the brain to 100 μM CGP54626, a selective GABA_B receptor antagonist, and 250 μM picrotoxin, a chloride channel blocker[52] and antagonist of GABA_A receptors (Fig. 7A). As expected, GABA blockers affected *wt* firing rates (Fig. 7B) but had no effect on PN spiking in *cac^RNAi* flies, where iLNs are hardly recruited (Fig. 6C). When blocking GABA receptors in *wt* flies, we observed a significant increase in the variability of PN activity but the temporal integration window of PN responses was unaffected (Fig. 7C). This suggests that iLNs only affect certain aspects of the code reliability. Predictably, blocking GABAergic transmission had no effect whatsoever on variability and only a minor effect on the temporal integration window in presynaptic *cac^RNAi* (Fig. 7C).

The AL also contains another population of inhibitory local neurons, which release glutamate. These interneurons mainly inhibit PNs, but were shown to have effectively similar roles as the GABAergic iLNs on the activity of the AL neural circuit[23]. Thus, the pharmacological experiments affect both the presynapse of ORNs and the postsynapse of PNs. Since iLNs mainly act by activating GABA_B receptors in the presynaptic terminal of ORNs[24,27], we tested whether driving *GABA-B-R2^RNAi* in ORNs (Fig. 7D) has similar effects on PN coding reliability as pharmacologically blocking the inhibitory iLNs (Fig. 7C). Indeed, *GABA-B-R2^RNAi* increased the variability at high firing rates but did not affect the temporal integration window (Fig. 7E). The glutamatergic iLNs inhibit PNs by activating the glutamate receptor GluClα[23]. Therefore, we expressed an RNAi construct against *gluClα* in PNs (Fig. 7F) and tested their odor response and coding reliability. Knockdown of the GluClα receptor in PNs had a strong effect on coding variability, as observed for the pharmacological experiments

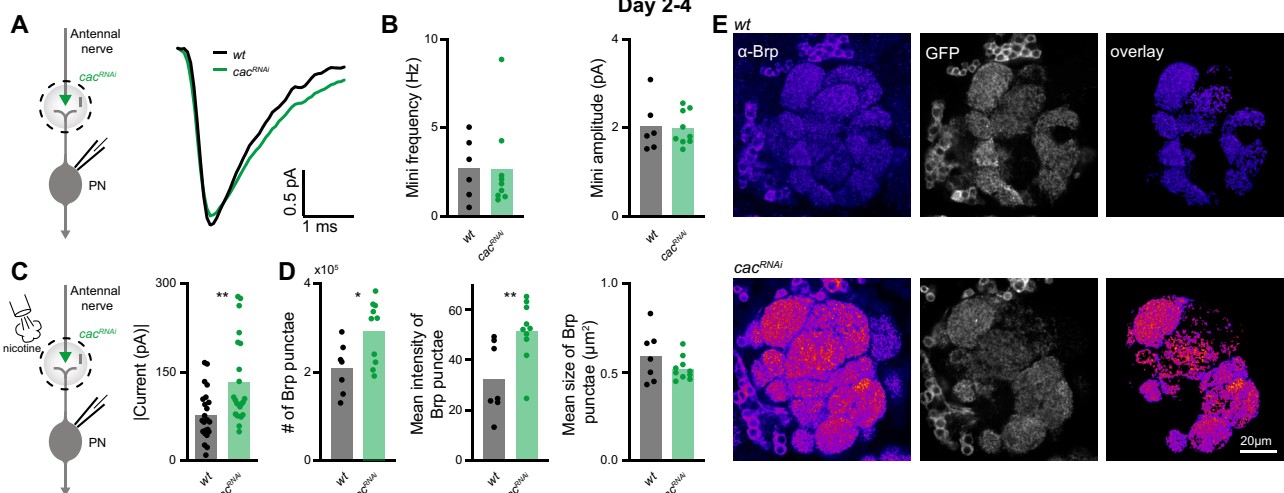

**Fig. 5 | A homeostatic increase in AZ numbers compensates for the drop in release probability. A** Left, experimental scheme. Postsynaptic responses to spontaneously occurring single-vesicle fusions (minis) were recorded in PNs in 2–4-day-old flies. Right, averaged minis in PNs of *wt* (black) and upon *cac* knockdown in ORNs (green). **B** Neither mini frequency (left) nor amplitude (right) differ significantly between genotypes (*wt*, $n = 6$; *cac^RNAi*, $n = 9$ flies). **C** Left, experimental scheme. Nicotine (100 μM) was applied to the AL and currents were measured in PNs. Right, Driving *cac^RNAi* with *Orco-GAL4* in ORNs increases the PN current amplitude in response to a nicotinic puff. Together with the unaltered minis, this indicates an increase in postsynaptic receptor fields. *GH146-QF* drove *QUAS-GFP* (*wt*, $n = 23$; *cac^RNAi*, $n = 22$ flies). Two-sided permutation test, $p = 0.002$. **D** Confocal microscopy shows a significant increase in the number and fluorescence intensity of individual Brp punctae at ORN-PN connections following *cac* knockdown. The average size of Brp punctae remained unaltered (*wt*, $n = 7$; *cac^RNAi*, $n = 10$ flies). Two-sided two sample *t*-test, $p = 0.02$ (# of Brp punctae), $p = 0.009$ (Mean intensity). **E** Example confocal images of an individual plane through the antennal lobe in control (upper panel; *orco-Gal4/+*; *GH-146-QF,QUAS-GFP/+*) and *cac^RNAi* flies (lower panel; *orco-Gal4/ UAS-cac^RNAi*; *GH146-QF,QUAS-GFP/+*) stained against Brp (fire). To restrict the analysis to ORN AZs with an excitatory PN as postsynaptic partner, the Brp signal was overlaid with masks generated by imaging GFP under control of *GH146* (gray). For all panels \*$p < 0.05$, \*\*$p < 0.01$, for detailed statistical analysis, see Table S1.

and by blocking the inhibitory presynaptic circuit (Fig. 7G). In addition, we also observed an overall decrease in correlation (Fig. 7G). However, in contrast to *cac^RNAi*, which increased PN integration time (Fig. 1D), we did not observe any such increase in *gluClα^RNAi*, if at all, the integration time of PNs was slightly decreased (Figs. 7G and S12). In summary, both pharmacological and genetic approaches suggest that the inhibitory AL circuit affects the variability of PN activity but not the temporal reliability.

**Reducing neural coding reliability disrupts behavioral reliability**
The above results demonstrate that reducing AZ $Ca^{2+}$ channel expression specifically in ORNs decreases the coding reliability by PNs. To examine whether these changes are relevant to behavioral output we tested the flies' ability to identify an ecologically relevant stimulus. To reduce response variability, which is often observed for naïve behavior[53,54], we paired isopentyl acetate with an electric shock, using a custom built apparatus[26,53–56] and tested the accuracy of the animals' behavioral responses, i.e. avoidance of isopentyl acetate. Following prolonged exposure (2 min) to the olfactory stimulus, presynaptic *cac^RNAi* animals were equally successful as their parental controls in classifying the odor (Fig. 8A, B), indicating no impairment in the flies' learning capabilities. However, when flies navigate in their natural habitat, olfactory stimuli are often brief and repetitive. Since such short stimuli could not be delivered in the above paradigm, we used an alternative assay in which the flies walked on a Styrofoam ball. This setting allowed us to present the same pattern of olfactory stimuli as used for the electrophysiological recordings (Fig. 1). Ten one-second trials of the shock-paired odor isopentyl acetate were presented and the response was classified as correct if flies turned away from the odor (Fig. 8C). Indeed, *cac^RNAi* flies made significantly fewer correct choices as a result of identifying isopentyl acetate less reliably (Fig. 8D). This reduction in correct stimulus classification was not due to a learning defect since the learning index was similar between *wt* and *cac^RNAi* animals (Fig. 8B). Furthermore, *cac* knockdown was only performed in

the first order ORNs and not in third order neurons, which are required for memory formation. A similar result was obtained with *syn^RNAi* (Fig. S13), supporting the notion that disrupted SV release and decreased neural code reliability reduce behavioral reliability. Since our electrophysiological results showed that homeostatic synaptic plasticity rescues the neural code reliability only at lower odor concentrations, we tested whether behavioral reliability was also rescued at low stimulation intensities. The overall learning scores of *wt* and *cac^RNAi* flies were similar at low (Fig. 8E) and high odor concentrations (Fig. 8B). As predicted, *cac^RNAi* flies were as successful as *wt* animals in identifying the low concentration odor (Fig. 8F). In contrast, while *cac^RNAi* animals displayed normal memory formation at day 0 (Fig. 8G), low concentration odor classification was significantly impaired in young flies (Fig. 8H). This matches our finding that homeostatic compensation takes a day to develop. In summary, these results demonstrate how synaptic plasticity can maintain neural coding reliability and enable the animal to consistently classify an important physiological stimulus.

## Discussion
Here we addressed two distinct yet related questions. First, we provide in vivo evidence that the molecular integrity of the AZ is important for neural coding reliability. Second, we demonstrate that homeostatic synaptic plasticity operates at the ORN-PN synapse and that this process compensates a reduction in release probability by increasing the number of ORN AZs. The homeostatic adjustment rescues eEPSC amplitudes in PNs and neural coding reliability. We further show that the restored neural reliability maintains behavioral performance. Finally, we demonstrate that the homeostatic compensation is limited in its efficiency and fails to maintain reliable neural coding and behavior at elevated firing rates in response to high odor concentrations.

Using in vivo recordings, we show that the high release probability of ORN AZs is required for a reliable neural code in postsynaptic PNs in response to physiological odor stimuli. Knockdown of *cac* in ORNs

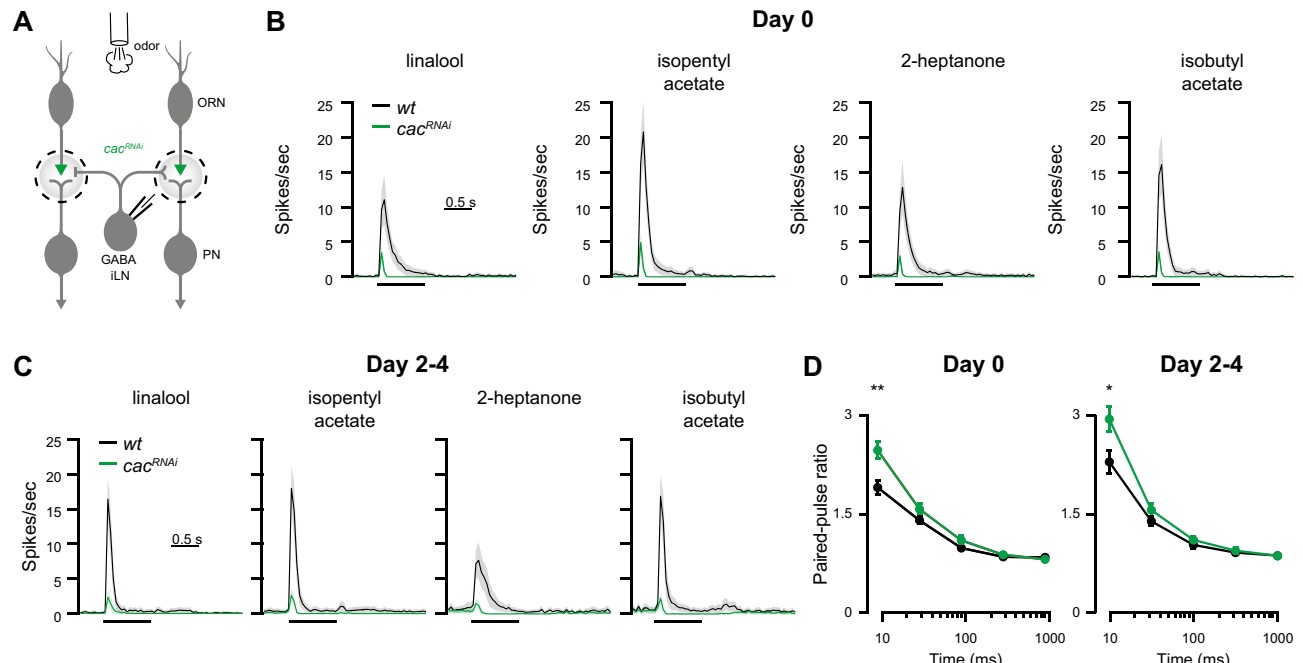

**Fig. 6 | Presynaptic *cac* knockdown abolishes iLN odor responses.**
**A** Experimental scheme. *cac^RNAi^* was driven in ORNs using *Orco-GAL4*. Whole-cell patch clamp recordings were made from iLNs labeled with the *449-QF* driver line. **B** PSTH of the iLN population response to four odors examined as indicated (shaded areas represent SEM, the odor pulse is labeled with a black bar) for 0-day-old flies. Knockdown of *cac* resulted in an almost complete abolishment of iLN odor responses (final odor dilution of $5 \times 10^{-2}$; $n = 49$–$50$ flies). **C** PSTH of the iLN population response to four odors examined as indicated (shaded areas represent SEM, the odor pulse is labeled with a black bar) for 2–4-day-old flies. Knockdown of *cac* resulted in an almost complete loss of iLN odor responses (final odor dilution of $5 \times 10^{-2}$; $n = 49$-$50$ flies). **D** Quantification of the paired-pulse ratio at different inter-stimulus intervals for 0 day (left) and 2–4-day-old (right) flies (10 ms, 30 ms, 100 ms, 300 ms, and 1000 ms). *cac^RNAi^* significantly increased paired-pulse facilitation at short inter-pulse interval [*wt* day 0, $n = 22$ (10, 30, 100, and 300 ms), $n = 21$ (1000 ms); *cac^RNAi^* day 0, $n = 28$ flies. *wt* day 2–4, $n = 13$; *cac^RNAi^* day 2–4, $n = 15$ (10 ms), $n = 16$ flies (30, 100, 300, and 1000 ms)]. Error bars represent SEM. Two-sided two sample *t*-test, $p = 0.002$ (10 ms day 0), $p = 0.025$ (10 ms day 2–4). For all panels **$p < 0.01$, for detailed statistical analysis, see Table S1.

---

reduced their release probability and affected neural code reliability in several ways. First, the high firing rates of the initial transient phase of PN odor responses became more variable due to reduced recruitment of iLNs. Second, the onset of odor responses was delayed and more variable. Third, the temporal dynamics of the odor response became less reliable. This third effect results from monosynaptic changes to ORN-PN transmission. At a behavioral level, we find that reduced coding reliability of PNs impairs the animals' ability to correctly classify a physiologically relevant stimulus.

We found that *cac^RNAi^* diminished neuronal coding reliability mainly at high firing rates. This was true for both the monosynaptic as well as the circuit mechanisms. In terms of the circuit effects on rate code reliability, this is in line with the recruitment of iLNs. iLNs sample many types of ORNs with different response profiles[41] and respond linearly to ORN activity[49]. Thus, iLNs respond most strongly to high ORN firing rates, which in turn are triggered by high odor intensities. Correspondingly, almost entirely eliminating the recruitment of iLNs by manipulating ORN AZs (Fig. 6), mostly affects processing of high firing rates. We find that a decrease in SV release probability increases the latency and jitter of eEPSC (Fig. 3), accompanied by an increase in the latency and jitter of postsynaptic action potential initiation (Fig. 1). We further show that this drop in the temporal precision of synaptic transmission increases in a frequency-dependent manner. This finding helps to explain why disrupted AZ function mainly impairs temporal coding at high firing rates and strong odor intensities.

The results reported here are based on a detailed analysis following the specific loss of Cac from ORN AZs. In principle, it is possible that the reduced reliability of neural coding and behavior could involve further Cac-dependent processes in addition to the direct control of SV fusion. However, our main findings also hold true for a further genetic manipulation of the presynapse. Knockdown of *syn* in ORNs disrupted neural coding and behavioral reliability in a similar manner to *cac^RNAi^* (Figs. S7, S8, and S13). This further supports the conclusion that impaired neurotransmitter release from ORNs is causative for the observed phenotypes at network and behavioral levels. It should be noted that the homeostatic compensation induced by *cac^RNAi^* expression prevents a simple interpretation of how Cac and neurotransmitter release probability shape neural coding under normal physiological conditions. In future work, it would therefore be of interest to test how an acute disruption of *cac* affects the circuit, e.g. by employing cell-specific tools based on a temperature-sensitive mutant[28,57].

Theoretical studies have suggested that decreasing SV release probability should reduce neural coding reliability[12–15,58–61], although exceptions have also been demonstrated[62,63]. However, whether these theoretical studies hold true in vivo, with highly dynamic synaptic properties, in complex neural circuits, and in response to physiological stimuli has remained an open question. The present study provides in vivo evidence that interfering with the molecular control of SV release results in reduced neural coding reliability. Notably, the ORN-PN synapse is surprisingly resilient to reduced AZ $Ca^{2+}$ channel expression and compensates the drop in release probability by increasing the number of AZs to yield normal eEPSC amplitudes. Homeostatic synaptic plasticity features at the *Drosophila* NMJ[8,9], mushroom body[64], and also operates in the antennal lobe to match synaptic strength to PN excitability[21,65]. Interestingly, *cac^RNAi^* decreases synaptic transmission to a greater extent from ORNs to iLNs than from ORNs to PNs (compare Figs. 1C and 6C). Thus, the functional impact of Cac or mechanisms to compensate for reduced $Ca^{2+}$ channel expression differ between ORN AZs depending on the identity of the

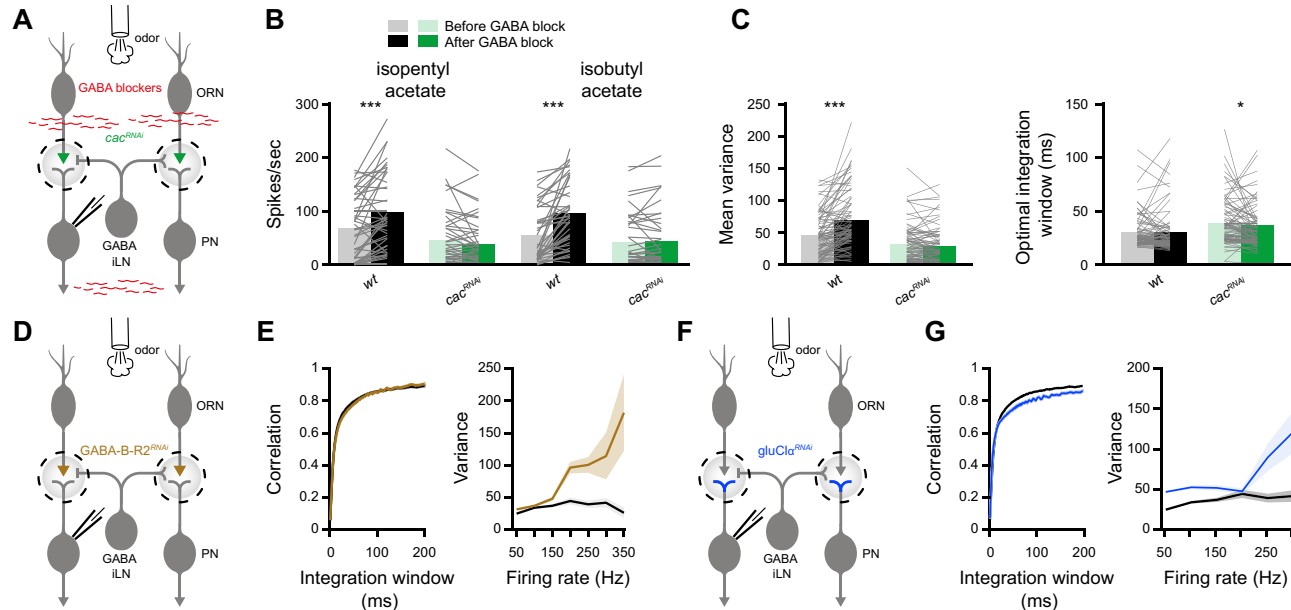

**Fig. 7 | Monosynaptic effects underlie temporal reliability whereas circuit effects underlie rate code reliability. A** Experimental scheme. GABA receptors blockers were applied by bath perfusion. Whole-cell patch clamp recordings were made from PNs labeled with the *GH146-QF* driver line. *UAS-cac$^{RNAi}$* was expressed in ORNs using *Orco-GAL4*. **B** Mean firing rate during the odor response. Application of GABA blockers significantly increased *wt* firing rates but had no effect on *cac$^{RNAi}$*. [*wt n* = 45 (isopentyl acetate), *n* = 43 (isobutyl acetate); *cac$^{RNAi}$ n* = 46 (isopentyl acetate), *n* = 44 flies (isobutyl acetate)]. Two-sided permutation test, *p* < 0.001 (*wt*, isopentyl acetate), *p* < 0.001 (*wt*, isobutyl acetate). **C** Left, temporal integration window and right, firing-rate reliability analysis for *wt* and *cac$^{RNAi}$* flies before and after the application of GABA receptors blockers. The data were pooled for the presentation of two odors (isopentyl acetate and isobutyl acetate) and across all firing-rates (*wt n* = 88, *cac$^{RNAi}$ n* = 90 flies). Two-sided permutation test, *p* < 0.001 (variance, *wt*), *p* = 0.041 (integration window, *cac$^{RNAi}$*). **D** Experimental scheme. An

RNAi construct against the GABA-B-R2 receptor was expressed in ORNs using *Orco-GAL4*. Whole-cell patch clamp recordings were made from PNs labeled with the *GH146-QF* driver line. **E** Left, temporal integration window and right, firing-rate reliability analysis for *wt* and *GABA-B-R2$^{RNAi}$* flies (brown). The data were pooled for the presentation of two odors (isopentyl acetate and isobutyl acetate) and across all firing-rates (*n* = 50–72 flies). Error bands represent SEM. **F** Experimental scheme. An RNAi construct against the GluClα receptor was expressed in PNs using *GH146-GAL4*. Whole-cell patch clamp recordings were made from PNs. **G** Left, temporal integration window and right, firing-rate reliability analysis for *wt* and *gluClα$^{RNAi}$* flies (blue). The data were pooled for the presentation of two odors (isopentyl acetate and isobutyl acetate) and across all firing-rates (*n* = 50-72 flies). Error bands represent SEM. For all panels **p* < 0.05, ***p* < 0.001, for detailed statistical analysis, see Table S1.

postsynaptic partner. Our imaging data indicate that in fact two compensatory plasticity processes are triggered by *cac$^{RNAi}$*: (i) At both ORN-PN and ORN-iLN synapses the mean size and fluorescence intensity of individual Brp punctae is significantly increased (Figs. 5 and S11). This likely reflects higher Brp levels or more densely packed protein[10] at individual AZs. As Brp has been shown to cluster Cac at release sites, increasing local Brp concentrations at AZs may help to ensure transmitter release despite a limited number of remaining Ca$^{2+}$channels, e.g. by optimizing SV-channel coupling distance[47]. Interestingly, the increased local Brp concentration may well be related to its role at the larval *Drosophila* NMJ, where Brp serves as a substrate for presynaptic homeostatic plasticity together with Cac[8–10,66,67]. Our data suggest that Brp performs a related function in the antennal lobe and demonstrate that Cac can serve as both a substrate and a trigger for homeostatic synaptic plasticity. (ii) The number of synapses only increases at ORN-PN connections. This is visualized by an increase in the number of Brp punctae following *cac* knockdown (Fig. 5). Moreover, the electrophysiological analyses indicate that the additional presynaptic sites are matched by a corresponding increase in functional postsynaptic partners (Fig. 5A–C). It is currently unclear what signaling pathway instructs the target-specific formation of additional synapses and why such a compensatory increase in synapse numbers does not occur at ORN-iLN connections (Figs. 6 and S11). Interestingly, a recent study demonstrated that during the time window of homeostatic compensation, i.e. the first day post-eclosion, ORNs and PNs show structural changes involving increased Brp expression and neurite expansion, while iLNs do not[68]. It is conceivable that this developmental stability prevents or limits structural

homeostatic plasticity at ORN-iLN synapses. Moreover, additional parallels may exist with the NMJ, where the expression of presynaptic homeostatic plasticity is compartmentalized to a subset of a motoneuron's synapses depending on their target muscle[69]. These considerations emphasize the importance of further future studies on the molecular heterogeneity of AZs in the olfactory system[70,71] and of elucidating the physiological properties of individual synapses in the context of neural information processing.

## Methods

### Fly strains

Fly strains (see below) were raised on cornmeal agar under a 12 h light/12 h dark cycle at 25 °C. The following fly strains were used: *UAS-cac$^{RNAi}$*[72] (VDRC ID 5551), *UAS-syn$^{RNAi}$* (BDSC_82983), *UAS-gluClα$^{RNAi}$* (BDSC_53356), *UAS-GABA-B-R2$^{RNAi}$* (BDSC_50608), *GH146-QF,QUAS-mCD8-GFP* (BDSC_30038), *Orco-GAL4* (BDSC_26818), *UAS-GCaMP6f* (BDSC_52869), *449-QF*[51], *QUAS-mCD8-GFP* (BDSC_30002).

For electrophysiology and structural imaging experiments *wt* flies were *Orco-Gal4; GH146-QF, QUAS-GFP* or *Orco-Gal4/ QUAS-GFP; 449-QF/+*.

### Olfactory stimulation

Odors (purest level available) were obtained from Sigma-Aldrich (Rehovot, Israel). Odor flow of 0.4 l/min (10$^{-1}$ or 10$^{-3}$ dilution) was combined with a carrier air stream of 0.4 l/min using mass-flow controllers (Sensirion) and software controlled solenoid valves (The Lee Company). This resulted in a final odor dilution of $5 \times 10^{-2}$ or $5 \times 10^{-4}$ delivered to the fly. Odor flow was delivered through a 1/16 inch ultra-

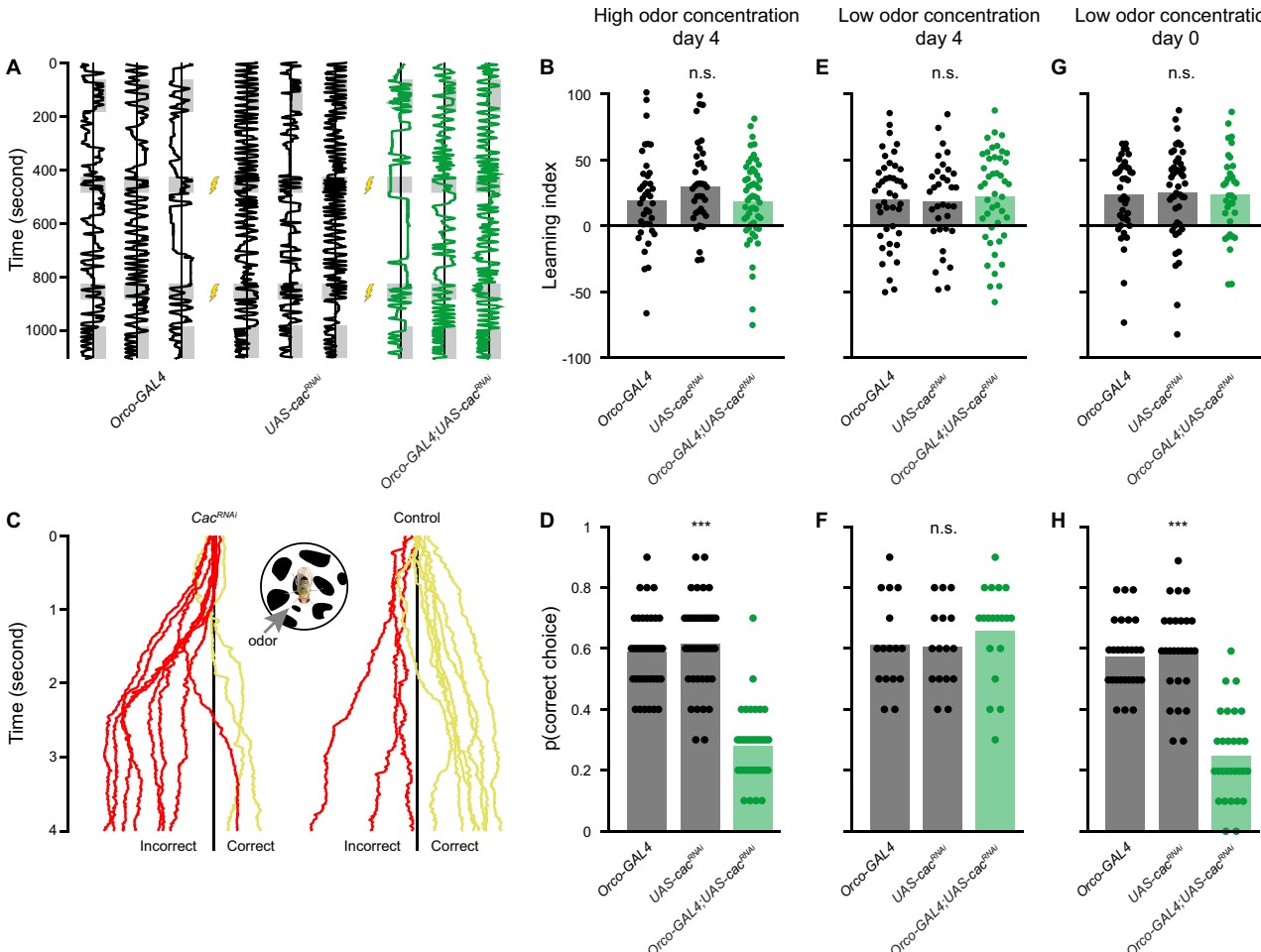

**Fig. 8 | Knockdown of *cac* reduces behavioral reliability. A** Experimental scheme for the learning paradigm. Flies were constrained in a linear chamber with isopentyl acetate presented on one side (gray). For pairing with an electric shock, the odor was presented on both sides of the chamber. Examples of single fly trajectories are shown. **B** Learning performance for prolonged odor exposure at high odor concentration. No significant differences in the learning index were observed between the parental controls and *cac^RNAi* in ORNs. Each dot represents a single fly [*n* = 40 (*Orco-GAL4*), *n* = 37 (*UAS-cac^RNAi*), *n* = 54 (*Orco-GAL4;UAS-cac^RNAi*) flies]. **C** Trained flies were tested in an assay where flies walked on a ball and the shock-paired odor was presented from the indicated side. Ten one-second odor trials were presented as in Fig. 1. Examples for single fly trajectories along the virtual *X*-axis are presented. Choice was classified as correct (yellow) if the mean *X*-axis position following the odor presentation was rightward of the *X*-axis midline (black line) and as incorrect (red) otherwise. **D** The behavioral reliability measure was defined as the probability of correctly classifying the shock-paired odor. Knockdown of *cac* in ORNs reduces the behavioral reliability compared to the parental controls at high odor concentration. Each dot represents a single fly [*n* = 39 (*Orco-GAL4*), *n* = 40 (*UAS-cac^RNAi*), *n* = 36 flies (*Orco-GAL4;UAS-cac^RNAi*)]. Two-sided permutation test, *p* < 0.001 with tukey-kramer correction for multiple comparisons. **E** Learning performance for

prolonged odor exposure at low odor concentration in 4-day-old flies. No significant differences in the learning index were observed between the parental controls and *cac^RNAi* in ORNs. Each dot represents a single fly [*n* = 43 (*Orco-GAL4*), *n* = 34 (*UAS-cac^RNAi*), *n* = 45 flies (*Orco-GAL4;UAS-cac^RNAi*)]. **F** Behavioral reliability measure as in panel D for low odor concentration in 4-day-old flies. Knockdown of *cac* in ORNs did not affect the behavioral reliability compared to the parental controls at low odor concentration. Each dot represents a single fly [*n* = 16 (*Orco-GAL4*), *n* = 16 (*UAS-cac^RNAi*), *n* = 19 flies (*Orco-GAL4;UAS-cac^RNAi*)]. **G** Learning performance for prolonged odor exposure at low odor concentration in 0-day-old flies. No significant differences in the learning index were observed between the parental controls and *cac^RNAi* in ORNs. Each dot represents a single fly [*n* = 40 (*Orco-GAL4*), *n* = 47 (*UAS-cac^RNAi*), *n* = 34 flies (*Orco-GAL4;UAS-cac^RNAi*)]. **H** Behavioral reliability measure for low odor concentration in 0-day-old flies. Knockdown of *cac* in ORNs reduces the behavioral reliability compared to the parental controls at low odor concentration, consistent with the absence of homeostatic compensation. Each dot represents a single fly [*n* = 27 (*Orco-GAL4*), *n* = 27 (*UAS-cac^RNAi*), *n* = 29 flies (*Orco-GAL4;UAS-cac^RNAi*)]. Two-sided permutation test, *p* < 0.001 with tukey-kramer correction for multiple comparisons. For all panels \*\*\**p* < 0.001, for detailed statistical analysis, see Table S1.

chemical-resistant Versilon PVC tubing (Saint-Gobain, NJ, USA) placed 5 mm from the fly's antenna.

### Electrophysiology

Female flies were anesthetized for 1 min on ice. A single fly was fixed to aluminum foil using wax. To expose the brain, the cuticle and trachea were removed. The brain was superfused with carbonated solution (95% $O_2$, 5% $CO_2$) containing (in mM): 103 NaCl, 3 KCl, 5 trehalose, 10 glucose, 26 $NaHCO_3$, 1 $NaH_2PO_4$, 1.5 $CaCl_2$, 4 $MgCl_2$, 5 N-Tris (TES), pH adjusted to 7.3. For in vivo whole-cell recordings 0,1 or 2–4-day-old flies were used as previously described[26]. Briefly, the brain was

visualized on a Scientifica Slice Scope with a ×40 water immersion objective. Patch pipettes of 9–12 MΩ resistance were used. The intracellular solution contained (in mM): potassium aspartate 140, HEPES 10, KCl 1, MgATP 4, $Na_3GTP$ 0.5, EGTA 1, pH adjusted to 7.3 and osmolarity 265 mOsm. PNs were randomly patched from both the lateral and medial cluster and iLNs were randomly patched from the lateral cluster. The recordings were performed using an Axon Instruments MultiClamp 700B amplifier and pClamp 10.7 in current- or voltage-clamp mode. Data was low-pass filtered at 1 kHz and sampled at 50 kHz. Upon break-in to the cell, a small constant current was applied to maintain a membrane potential of −60 mV. Action

potentials were extracted using a custom MATLAB code followed by manual inspection to verify correct identification. Spike times were aligned to the beginning of the rising phase of the membrane potential, indicating the beginning of the odor stimulus. In experiments where GABA blockers were used, 100 μM of CGP 54626 (Tocris, CAS: 149184-21-4) and 250 μM of picrotoxin (Sigma-Aldrich, CAS: 124-87-8) were bath applied. eEPSCs were evoked by stimulating ORN axons with a minimal stimulation protocol via a suction electrode[25]. Brief pulses (typically 50 μs) were passed through the innervating nerve using a constant current stimulator (Digitimer, DS3 Isolated Current Stimulator). To measure eEPSC amplitudes and kinetics, 32 eEPSCs were evoked at 1 Hz and averaged. eEPSC latency was measured from the beginning of the stimulation artifact to the beginning of the eEPSC rising phase. Paired-pulse recordings were made at 0.2 Hz with interstimulus intervals of (in ms): 10, 30, 100, 300, and 1000. For each interval 20 traces were averaged. Ten seconds of rest were afforded to the cell in between recordings. The amplitude of the second response in 10 ms inter-pulse recordings was measured from the peak to the point of interception with the extrapolated first response. Data were analyzed using MATLAB.

For in vivo patch-clamp recordings of minis, electrodes (6–8 MOhm) were filled with internal solution containing (in mM): Potassium aspartate 125, $CaCl_2$ 0.1, EGTA 1.1, HEPES 10, MgATP 4, $Na_3GTP$ 0.5, pH adjusted to 7.3 and osmolarity 265 mOsm. To visualize PN morphology, Biocytin (3 mg/ml) was added to the intracellular solution. Throughout measurements, preparations were perfused with oxygenated external saline, containing (in mM): NaCl 103, KCl 3, $CaCl_2$ 1.5, $MgCl_2$ 4, $NaH_2PO_4$ 1, $NaHCO_3$ 26, TES 5, Trehalose 10, Glucose 10, pH adjusted to 7.3 and 277 ± 2 mOsm. Additionally, 4 μM TTX (Carl Roth, 6973) was supplied to the bath solution to suppress spontaneous spiking. For recordings, PNs were held at a command potential of −80 to −100 mV. Signals were low-pass filtered at 1 kHz and digitized at 10 kHz. Mini detection and analysis was carried out with ClampFit 11.0.3, Molecular Devices.

## Pharmacology

The following drugs were used: nicotine (Sigma-Aldrich #N3876) and TTX (Alomone Labs #T-550). In all cases, stock solutions were prepared were diluted to the final concentration before experiments. Drugs were applied either by bath application or were injected directly to the AL using a pico-injector (Harvard Apparatus, PLI-100).

## Functional imaging

Imaging was carried out as previously described[54,56,73] using two-photon laser-scanning microscopy (DF-Scope installed on an Olympus BX51WI microscope). Female flies were anesthetized on ice and then moved to a custom built chamber and fixed to aluminum foil using wax. Cuticle and trachea were removed from the area of interest and the exposed brain was superfused with carbonated solution as described above. Odors were delivered at a final dilution of $5 \times 10^{-2}$ and fluorescence was excited by a Ti-Sapphire laser (Mai Tai HP DS, 100 fs pulses) centered at 910 nm, attenuated by a Pockels cell (Conoptics) and coupled to a galvo-resonant scanner. Excitation light was focused by a ×20, 1.0 NA objective and emitted photons were detected by GaAsP photomultiplier tubes (Hamamatsu Photonics, H10770PA-40SEL), whose currents were amplified (Hamamatsu HC-130-INV) and transferred to the imaging computer (MScan 2.3.01). All two-photon imaging experiments were acquired at 30 Hz.

## Immunohistochemistry

Whole mount stainings were performed essentially as previously reported[73]. For Brp analysis, brains were fixed with 4% PFA for 2 h, washed with PBT (0.3%), and blocked overnight using 5% normal goat serum in PBT at 4 °C, before adding mouse-anti Brp (nc82, 1:50; provided by E. Buchner; RRID: AB_528108). Following another overnight

incubation at 4 °C, samples were washed with PBT before applying goat anti-mouse STAR RED (1:200; Abberior #2-0002-011-2, RRID:AB_2810982) overnight. After washing with PBT, the samples were stored in mounting medium (Abberior MOUNT, LIQUID ANTIFADE, #MM-2009). To preserve tissue morphology, imaging spacers (Sigma-Aldrich #GBL654008) were used. For verification of *cac* knockdown at ORN pre-synapses, we followed a previously published protocol[74]. Whole mounts were fixed with Bouins fixative (Carl Roth #6482.3), washed with PBS, and blocked in PBT (0.2%) with 5% NGS at 4 °C overnight. This was followed by an overnight incubation with rabbit anti-Cac (1:1000) in blocking solution, before the samples were washed several times with PBT. Subsequently, whole mounts were incubated with biotinylated goat anti rabbit IgG (1:500; Jackson ImmunoResearch Labs #111-065-003, RRID:AB_2337959), washed with PBT, and subsequently incubated with STAR RED streptavidin (1:500; Abberior #STRED-0120). Samples were stored in mounting medium (Abberior MOUNT, LIQUID ANTIFADE, #MM-2009) until final use.

## Confocal microscopy

Images were acquired with an Abberior INFINITY LINE system (upright Olympus BX63F) equipped with a ×60, 1.42 NA oil immersion objective. Laser settings were kept constant and image acquisition alternated between genotypes. Image analysis was done using ImageJ 1.5 3 C (National Institutes of Health, Bethesda) and the analysis was performed on each image in a stack. Data sets exceeding the maximal slice number of 125 slices per stack were excluded from the analysis. To restrict image analysis to signals at ORN-PN synapses, masks of the GFP signal (driven with *GH146*) were created and overlayed with the Brp or Cac channel. Individual punctae were detected with the "Find Maxima" command and quantified via "Analyze Particles".

## Behavioral chambers

Experiments were performed using a custom-built, fully automated apparatus[25,53,55,56]. Single male and female flies were housed in clear chambers (polycarbonate, length 50 mm, width 5 mm, height 1.3 mm). Mass flow controllers (CMOSens PerformanceLine, Sensirion) were used to control air flow. An odor stream (0.3 l/min) obtained by circulating the air flow through vials filled with a liquid odorant was combined with a carrier flow (2.7 l/min). Isopentyl acetate was prepared at $10^{-1}$ or $10^{-3}$ dilution. Fresh odors were prepared daily.

Two identical odor delivery systems were used each delivering Isopentyl acetate independently to each half of the chamber. The total flow (3 l/minute, carrier and odor stimulus) was split between 20 chambers. The air flow from the two halves of the chamber converged at a central choice zone. The 20 chambers were stacked in two columns each containing 10 chambers and were backlit by 940 nm LEDs (Vishay TSAL6400). Images were obtained by a MAKO CMOS camera (Allied Vision Technologies) equipped with a Computar M0814-MP2 lens. The apparatus was operated in a temperature controlled incubator (Panasonic MIR 154) at 25 °C.

Fly position was extracted from video images using a virtual instrument written in LabVIEW 7.1 (National Instruments). The same virtual instrument was also used to control odor delivery. Data were analyzed in MATLAB 2018a (The MathWorks). The conditioning protocol included 12 equally spaced 1.25 s electric shocks at 50 V, and was repeated twice. The learning index was calculated as (preference for isopentyl acetate before training) – (preference for isopentyl acetate after training). Flies that were used for the ball assay in Fig. 8C were given a training protocol that consisted of 12 equally spaced 1.25 s electric shocks at 50 V, repeated 6 times with 15 min interval. Flies were then tested on the ball assay 12 h later.

## Ball assay

Flies walked on a treadmill ball as previously described[75]. Briefly, the fly thorax was glued to a fine metal rod using wax in a manner that

allowed for free limb movement. Flies were then placed on a Styrofoam ball with a diameter of 9 mm which floated on an air-steam in a custom made ball holder. Videos were captured using a Blackfly S (FLIR® Systems) camera fitted with a computer macro zoom 0.3–1X, 1:4.5 lens at a frame rate of 100 FPS. The ball was illuminated by infrared LEDs and the fly position was tracked offline using FicTrac[76]. Odor delivery was performed as for the electrophysiological experiments described above.

### Quantification and statistical analysis

**Integration window analysis.** For this analysis, only the first two seconds after the beginning of the odor stimulus were used. Spike trains were then binned with bin sizes ranging from 1 ms to 200 ms with 1 ms intervals. For each bin size, the pairwise Pearson correlation was calculated for all combinations of the 10 trials for each odor. Trials that did not show any odor response were excluded from this analysis. The optimal integration window was defined by using the Youden's index[39], which is the point of maximal distance between the correlation vs. bin size curve and the diagonal between the minimum and maximum point of said curve. This index was calculated for each individual pairwise correlation as a function of bin size and pooled across neurons, odors, and flies. The Youden's index is a commonly used measure for the optimal cutoff point of a curve, such as the one obtained by plotting correlation vs. bin size[77].

**Variance analysis.** For each fly, the mean and variance of the 10 stimulus repetitions were calculated using time bins of 20 ms. The mean spike count and variance during a 20 ms time window were calculated. The variance was pooled across flies and neurons with the same mean spike count.

### Statistics and data analysis

All statistical testing and parameter extraction were done using custom MATLAB code (The MathWorks, Inc.) and PRISM 9.3.0. All details of statistical tests are given in Table S1. Significance was defined as a p-value smaller than 0.05 and all statistical tests were two-sided. Normality assumption was tested using the Shapiro-Wilk test (https://www.mathworks.com/matlabcentral/fileexchange/13964-shapiro-wilk-and-shapiro-francia-normality-tests). In cases where the normality assumption was violated a permutation test was used using the 'permutationTest' function in MATLAB (https://github.com/lrkrol/permutationTest).

Effect size was calculated with the Measures of Effect Size (MES) Toolbox (https://github.com/hhentschke/measures-of-effect-size-toolbox/blob/master/readme.md). Permutation test was used using the 'permutationTest' function in MATLAB (https://github.com/lrkrol/permutationTest).

For presentation, bar plots with dots were generated using the UnivarScatter MATLAB ToolBox (https://www.mathworks.com/matlabcentral/fileexchange/54243-univarscatter), raster plots were generated with the Flexible and Fast Spike Raster Plotting ToolBox (https://www.mathworks.com/matlabcentral/fileexchange/45671-flexible-and-fast-spike-raster-plotting) and the shadedErrorBar function (https://github.com/raacampbell/shadedErrorBar) for shaded errors on imaging traces.

### Reporting summary

Further information on research design is available in the Nature Portfolio Reporting Summary linked to this article.

## Data availability

The source data used to generate the figures in this manuscript are available upon publication in the following GitHub page: https://github.com/ParnasLab/Rozenfeld-et-al-2023. Source data are provided with this paper.

## Code availability

The code used to generate the figures in this manuscript is available upon publication in the following GitHub page: https://github.com/ParnasLab/Rozenfeld-et-al-2023.

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

## Acknowledgements

We thank B. Goettgens and N. Naumann for technical support and Dr. Ya-Hui Chou and the Bloomington Stock Center for fly strains. We also thank David Morton for kindly sharing the Cacophony antibody. This work was supported by the European Research Council (676844, M.P.) and the Deutsche Forschungsgemeinschaft (project numbers 430156010/SPP 2205 and 426503586/KFO 5001 to R.J.K. and 408264519 to M.P. and R.J.K.).

## Author contributions

E.R.: conceptualization, methodology, investigation, formal analysis, software, writing–review and editing, and visualization. N.E.: investigation, formal analysis, writing–review and editing, and visualization. J.E.M.: investigation, formal analysis, and visualization. R.J.K.: initiated the project, conceptualization, methodology, formal analysis, writing–original draft, writing–review and editing, supervision, and funding acquisition. M.P.: initiated the project, conceptualization, methodology, investigation, formal analysis, software, writing–original draft, writing–review and editing, visualization, supervision, and funding acquisition.

## Competing interests

The authors declare no competing interests.
