## [Peer Review File · Nature Communications]

Homeostatic Synaptic Plasticity Rescues Neural Coding ReliabilityREVIEWER COMMENTS

Reviewer #1 (Remarks to the Author):

Rozenfeld et al. take advantage of *Drosophila* genetic and electrophysiological tools to test the flexibility and reliability of neuronal coding in the olfactory system.

The main approach is to knock down message levels of cacophony (*cac*) in ORN neurons by RNAi. *cac* is a gene that encodes the alpha 1 subunit of CaV2-type voltage gated calcium channels. The authors use *cac* knockdown as a tool to disrupt neurotransmission from presynaptic ORNs and then to measure effects upon its postsynaptic partners, the PNs (Figs. 1, 2) and the modulatory iLNs (Fig. 3) when odors are presented – and then they conduct a behavioral assay in adult flies (Fig. 4).

Loss of *cac* gene function disrupts the reliability of the information transfer when considering the effect that ORNs have upon PNs. This variability applies in cases where a high odor concentration (which elicits a higher degree of firing) is presented in animals aged 2-4 days (Fig. 1). It also applies in all cases where newly eclosed animals are presented with odors of any concentration. The idea is that over the course of development, the animal can adjust to the loss of *cac* by retargeting activity back toward baseline levels, due to an unknown homeostatic mechanism. The authors find some visual evidence consistent with such a mechanism: increased *Brp* signal, apposed to PN targets (Fig. 2).

The authors go on to examine the broader circuit, testing the effects of *cac* knockdown on the function of the modulatory iLNs. Unlike with the PNs, they do not observe a similar compensatory mechanism, at least vis a vis spiking. Patch electrophysiology in the iLNs showed muted responses in all cases after *cac* knockdown (Figs. 3A-D). In fact, this *cac* knockdown was able to occlude GABAergic signaling effects that iLNs can have upon PNs. This suggests that whatever homeostatic mechanism is at play after *cac* knockdown, it is limited to the ORN-to-PN connection.

Interestingly, the behavior of the flies matches key aspects of the electrophysiology (Fig. 4).

The core results of this study are interesting. These results could provide a roadmap for others in the field to start to uncover and unravel forms of homeostatic plasticity that work to ensure reliable signal coding. The data will be interesting to variety of neuroscientists, including those who study who study circuitry, those who study behavior, and those who study synaptic plasticity and neurotransmission. There are several points that could use clarification or clean-up prior to publication.

Main Points

1. The main point for this reviewer is that most of the conclusions for the study rely on a single manipulation to impair neurotransmission in the ORN: *cac*RNAi, using the VDRC line 5551. It is good that the authors verified a reduced calcium signal when expressing this line. Moreover, this reviewer's lab has previously used VDRC 5551, and we found it to be a reliable reagent. There are no concerns that cacophony message is being knocked down with VDRC 5551.

Rather, the concern is that broad general principles about coding and reliability of information transfer are inferred downstream of a single manipulation. The generalization here is that impaired synaptic vesicle release causes the phenotypes (see Abstract and elsewhere). This is logical based on the data and based on a *cac* manipulation. However, it is also formally possible that there are other effects downstream of *cac* knockdown that are contributing to the phenotypes, independent of SV release (e.g., intracellular calcium signaling downstream of influx).

Any sort of additional genetic manipulation known to impair SV release would be a welcome addition to the study. Such a manipulation would likely match the behavior of the *cac*RNAi. Note for editors – I am not requesting a re-do of the entire study using a different manipulation. Alternatively, the conclusions could be stated in a parsimonious way that these results are specific to *cac* loss, with added Discussion about the possibilities of what that could mean.

2. The homeostatic increase in Brp punctae number and signal intensity (Figs. 2U-V) represents a plausible mechanism for a form of compensation that could be happening. Two questions arise from this observation:

One question is how precisely the authors think this proposed homeostatic system is working? Intuitively, increased Brp alone should not completely make up for a loss of Cac. The synapse still needs calcium influx, and the *cac* gene is still knocked down. If, however, the increased Brp were restoring the calcium signal (for example, by recruiting whatever wild-type residual Cac remained at the synapse), that might help to explain the conundrum.

The second question deals with the Brp signal. The authors show quite nicely that the Brp signal in the ORNs that is apposed to the PNs increases (Fig. 2U-V). The authors go on to state by electrophysiology that the ORN-to-iLN connection undergoes no such potentiation over development. But they do not report Brp levels in ORNs apposed to the iLN postsynapse.

If it were the case that those connections are not potentiated after cac knockdown, then one should expect no increases in Brp signal. However, if one did see an increase in Brp signal apposed to iLNs, then that would call into question the homeostatic mechanism proposed for the ORN-to-PN connection. In that scenario, the Brp signal would seem to be correlative, not causative.

3. The variability after the knockdown of cac is interesting. To this reviewer, it seems to reflect a high variability that one sees in neurotransmission at low probabilities of release. In that sense, if one looks at the high odor concentration data (Fig. 1F) vs. the low odor concentration data (Fig. 1O) – just for wild type – one sees a much higher variance for the low concentration. So, it might not simply be the case that the system is broken at the high concentration and fixed at the low concentration. This reviewer wonders if an even higher odor concentration were presented in for 1F, would that ameliorate the difference observed? Alternatively, if a higher calcium concentration were used in the recording saline, would that be enough to erase the variance difference?

Side note: even though the average eEPSCs are not changed in Fig. 2B the cacRNAi data look more highly variable than control.

Minor Points

1. For genetic manipulations, cac (or UAS-cacRNAi) is more correctly italicized and lowercase. For Cacophony protein, uppercase Cac is correct. I know this is the kind of minor point that could end up being a time-consuming thing to fix because it would involve changing multiple pieces of text in every single figure. So, to be clear, this is a suggestion for an edit – not a required edit that changes anything about the clarity of the paper or figures.

2. The experiments testing 0 days vs. 2-4 are very helpful in terms of defining a temporal component to the plasticity. Do the authors happen to know when exactly the ORN-specific Gal4 turns on? That could indicate how long the animals were experiencing a loss of cac expression.

3. Figure 2U: I do not think the authors mean square meters (m^2). Square microns (μm^2) maybe?

4. The authors note in the Introduction and Discussion how forms of homeostatic plasticity have been characterized at synapses like the NMJ. Of interest to this study, it is known from that NMJ work that Cacophony is needed to promote homeostatic potentiation of release properties. On one level, that would seem to suggest an interesting connection to the current work. However, what is happening here seems different. In the present study, it seems like the loss of cac is the initial challenge to this circuit, not a substrate for compensation. At the NMJ, Cac appears to be a substrate of homeostatic plasticity

signaling machinery after a different type of challenge. Linking the two, Brp seems to act like a shared substrate of compensation in both systems.

Reviewer #2 (Remarks to the Author):

In this study the authors tested the impact of modulating pre-synaptic machinery of the olfactory receptor neurons (ORNs) on the temporal and rate coding of the olfactory stimuli in fruitfly antennal lobe projection neurons (PN). Overall, they observed that modulating pre-synaptic machinery reduces the PN odor response amplitude and increase the temporal variability of spike times, and jitter. They also went further and dissect these effects and found out that while rate code effects are mainly through multi-synaptic ORN to LN to PN synapse, the temporal code effects are monosynaptic for the ORN to PN synapse. Authors even dive deeper and identified the synaptic mechanisms of how GABAergic and Glutamtergic LNs can contribute to response reliability. Finally they tested the effects of these manipulations on olfactory behaviors and showed that while there were no differences in long odor exposure assays, when the odor exposure periods are reduced, animals with perturbed ORN pre-synaptic machinery performed significantly worse.

I think this is a technically well executed study, and very well written and easy to read manuscript. The authors did so many experiments that I had in mind, and as I read the manuscript, I felt like many of my questions are answered. I also sensed that the completeness of this nice story is partly due to some pervious rounds of revisions, which dig further into the mechanisms of the observed effects. Hence I support the publication of this manuscript, and I have only minor comments below, which I hope that it will help authors to clarify few things.

1) In general, the entire study relies on population effects in unidentified PNs. While I understand that it will be very difficult to use the Gal4 system both for identifying PNS and manipulating ORNs, is there no pharmacology that could target this pre-synaptic machinery while identifying a single type of stereotyped PN, and conclusively show the effect on spike reliability in genetically identified PNs?

2) Authors discuss a lot about trial to trial variability, and its effect in odor coding. A recent study in fruitfly PNs (<https://doi.org/10.1016/j.celrep.2021.110165>) have shown that repeated odor exposure increases reliability of odor representations and reduced ongoing spontaneous activity. It might be interesting to see whether the authors observe such increased reliability after repeated exposure depends on the AZ mechanism that is perturb in this paper.

3) Authors uses correlations as a measure for similarity but not Euclidean distance, could this help to better interpret the amplitude effects on the odor responses ?

- 4) Alternatively cosine distances can be a reliable measure for not amplitude dependent reliability across stimulus delivery. I think that the authors should test their reliability analysis by using multiple complementary measurements of reliability.
- 5) Spike reliability probably a lot more important for low odor concentrations, isn't it ? why the effect is more prominent as high concentrations ? does this have anything to the way the reliability is analyzed ?
- 6) In general it will help a lot to give a little more information in the main text about the quantifications, since it is not easy to go back and forth between legends and the main text to understand what exactly the authors did.
- 7) I also found some of the figures are not well arranged or labelled, some polishing of figures could make it easier to read this manuscript
- 8) Markus Meister lab have investigated encoding of visual information based on the timing of the first spike of retinal ganglion cells. The authors touch up on some of these concepts in this manuscript. I think it is certainly worth to cite these previous studies
- 9) Analysis in Figure 1E is not well referenced or explained in the text nor in the methods. Please try to clarify better.
- 10) Figure 1F, please plot the time bins in ms, not in number of bins.
- 11) Figure 1L-O, is there a significant increase in correlations in AZ perturbed animals ? not discussed ?
- 12) is there a significant effect in Figure 3K, I can not find any statistics

Reviewer #3 (Remarks to the Author):

This study defines *in vivo* roles for presynaptic calcium channels, presynaptic release probability and homeostatic mechanisms in neural coding reliability. This is achieved through cell type-specific genetic manipulation and rigorous electrophysiological analysis of a well-defined neural circuit within the *Drosophila* CNS. The work is convincing and impressive. It builds on previous studies to establish an attractive model for *in vivo* genetic analysis of neural circuits in the context of behavior, and the results include several novel and noteworthy findings of interest to the broader neuroscience community.

Strengths:

- 1) Approach. Despite significant technical challenges, the authors combine cell type-specific genetic manipulation with patch clamp recording and imaging of identified cell-types within a defined neural circuit. Moreover, examining odor responses at the circuit and behavioral levels provides a compelling *in vivo* physiological context for this work.

2) Key findings. KD of cac resulted in a combination of functional and compensatory morphological changes within the olfactory circuit. This might have been viewed as confounding, but the authors dug in and addressed these changes in a systematic way. In the process, they made noteworthy discoveries, including a first in vivo look at how release probability and homeostatic mechanisms can influence neural coding.

cac KD produced changes in coding reliability within the olfactory circuit. After these were characterized in the context of two distinct components, variability and temporal reliability, the underlying mechanisms were examined at the synaptic and circuit levels. cac KD did not change EPSC amplitude, however the authors noted an increase in facilitation, which is suggestive of decreased release probability. Accordingly, they considered that normal EPSC amplitudes may result from a compensatory change in synapse number or quantal size. Analysis of mEPSCs and immunocytochemical studies confirmed the former possibility. This thoughtful and systematic approach demonstrated homeostatic plasticity within the circuit and presented an opportunity to examine its role in coding reliability. Further analysis examined the different cell and synapse types within the circuit and their contributions using genetic and pharmacological methods. Although these are complex and challenging problems, even in a simple circuit, the authors addressed them experimentally and made a solid case for their interpretations. In the end, this study provides a detailed dissection of how synaptic mechanisms, circuit interactions and homeostatic plasticity can influence the components of coding reliability in distinct ways.

3) Model. Another strength of this study is its further development of an attractive model for in vivo genetic analysis of homeostatic mechanisms in the CNS. Studies in this model can address homeostatic plasticity within a defined neural circuit and physiological context and, importantly, in a fully developed, mature organism with established synaptic and circuit architecture.

4) The manuscript is clear and well-organized. The authors' efforts to provide a conceptual framework for understanding technical aspects of their work, including analysis of coding reliability, will be helpful to a general audience. The results are presented with a narrative that conveys the rationale for each step in their analysis. Finally, the authors emphasize complementary findings from different experimental approaches, and this further strengthens their conclusions.

Weaknesses (none) and minor suggestion:

This careful study reports complexities in genetic analysis that are easily overlooked or neglected. Homeostatic compensation for cac KD, and the resulting changes in synapse and circuit morphology, led to important findings in this study and cannot be considered a weakness. However, it does preclude simple conclusions about the specific role of cac and presynaptic release probability in neural coding under normal conditions. This caveat might be acknowledged in the discussion. It would be of interest if future studies in this model can examine the acute physiological consequences of disrupting cac and

presynaptic release probability. If feasible, this might be achieved by developing cell type-specific tools based on an existing temperature-sensitive cac mutant.

Rick Ordway

Response to reviewers, manuscript number NCOMMS-22-36913-T

General

We would like to thank the reviewers for their thoughtful comments, which have led to a significant improvement in the quality of our study.

Reviewer #1

Rozenfeld et al. take advantage of Drosophila genetic and electrophysiological tools to test the flexibility and reliability of neuronal coding in the olfactory system.

The main approach is to knock down message levels of cacophony (cac) in ORN neurons by RNAi. cac is a gene that encodes the alpha 1 subunit of CaV2-type voltage gated calcium channels. The authors use cac knockdown as a tool to disrupt neurotransmission from presynaptic ORNs and then to measure effects upon its postsynaptic partners, the PNs (Figs. 1, 2) and the modulatory iLNs (Fig. 3) when odors are presented – and then they conduct a behavioral assay in adult flies (Fig. 4).

Loss of cac gene function disrupts the reliability of the information transfer when considering the effect that ORNs have upon PNs. This variability applies in cases where a high odor concentration (which elicits a higher degree of firing) is presented in animals aged 2-4 days (Fig. 1). It also applies in all cases where newly eclosed animals are presented with odors of any concentration. The idea is that over the course of development, the animal can adjust to the loss of cac by retargeting activity back toward baseline levels, due to an unknown homeostatic mechanism. The authors find some visual evidence consistent with such a mechanism: increased Brp signal, apposed to PN targets (Fig. 2).

The authors go on to examine the broader circuit, testing the effects of cac knockdown on the function of the modulatory iLNs. Unlike with the PNs, they do not observe a similar compensatory mechanism, at least vis a vis spiking. Patch electrophysiology in the iLNs showed muted responses in all cases after cac knockdown (Figs. 3A-D). In fact, this cac knockdown was able to occlude GABAergic signaling effects that iLNs can have upon PNs. This suggests that whatever homeostatic mechanism is at play after cac knockdown, it is limited to the ORN-to-PN connection.

Interestingly, the behavior of the flies matches key aspects of the electrophysiology (Fig. 4).

The core results of this study are interesting. These results could provide a roadmap for others in the field to start to uncover and unravel forms of homeostatic plasticity that work to ensure reliable signal coding. The data will be interesting to variety of neuroscientists, including those who study who study circuitry, those who study behavior, and those who

study synaptic plasticity and neurotransmission. There are several points that could use clarification or clean-up prior to publication.

Main Points

1. The main point for this reviewer is that most of the conclusions for the study rely on a single manipulation to impair neurotransmission in the ORN: cacRNAi, using the VDRC line 5551. It is good that the authors verified a reduced calcium signal when expressing this line. Moreover, this reviewer's lab has previously used VDRC 5551, and we found it to be a reliable reagent. There are no concerns that cacophony message is being knocked down with VDRC 5551.

Rather, the concern is that broad general principles about coding and reliability of information transfer are inferred downstream of a single manipulation. The generalization here is that impaired synaptic vesicle release causes the phenotypes (see Abstract and elsewhere). This is logical based on the data and based on a cac manipulation. However, it is also formally possible that there are other effects downstream of cac knockdown that are contributing to the phenotypes, independent of SV release (e.g., intracellular calcium signaling downstream of influx).

Any sort of additional genetic manipulation known to impair SV release would be a welcome addition to the study. Such a manipulation would likely match the behavior of the cacRNAi. Note for editors – I am not requesting a re-do of the entire study using a different manipulation. Alternatively, the conclusions could be stated in a parsimonious way that these results are specific to cac loss, with added Discussion about the possibilities of what that could mean.

We thank the reviewer for this important suggestion. We have now added several experiments to the manuscript demonstrating that our main findings also hold true for a further genetic manipulation. When Synapsin (Syn), a presynaptic protein involved in SV release, is knocked down, the reliability of neuronal coding by PNs and behavioral reliability are both impaired in a similar manner to *cac^{RNAi}*. These results are presented in the new Supplemental Figures S7, S8 and S13.

We have also added a paragraph to the discussion pointing out that our results mainly relay on the loss of Cac and that the reduced reliability of neural coding and behavior could, in principle, involve further processes in addition to reduced SV release probability. Lines 382-393.

2. The homeostatic increase in Brp punctae number and signal intensity (Figs. 2U-V) represents a plausible mechanism for a form of compensation that could be happening. Two questions arise from this observation:

One question is how precisely the authors think this proposed homeostatic system is working? Intuitively, increased Brp alone should not completely make up for a loss of Cac. The synapse still needs calcium influx, and the cac gene is still knocked down. If, however, the increased Brp were restoring the calcium signal (for example, by recruiting whatever wild-type residual Cac remained at the synapse), that might help to explain the conundrum.

We thank the reviewer for drawing our attention to this intriguing point. We have added an experiment to directly quantify whether Ca²⁺ levels are changed following the homeostatic compensation. To this end, we measured the Ca²⁺ signal at day 0 in *cac^{RNAi}* flies and show that it is very similar to 4 day-old flies. Thus, the homeostatic compensation does not appear to involve an increase in the recruitment of Cac or other Ca²⁺channel types to ORN active zones (AZs). This data is presented in the new Supplemental Figure S10.

These results are consistent with a redistribution of Cac across a greater number of AZs, marked by Brp, during homeostatic plasticity. The increased fluorescence intensity of individual Brp punctae (Figure 4) likely reflects higher Brp levels (or more densely packed protein, see (Mrestani et al., 2021) at individual AZs (see also next question). As Brp has been shown to cluster Cac at release sites (Kittel et al., 2006), increasing local Brp concentrations at AZs may help to ensure SV release despite a limited number of remaining Ca²⁺channels, e.g. by optimizing SV-channel coupling distance. As a side note, the release probability appears to drop slightly from day 0 to day 1 or day 2-4, reflected by the further increase in PPR. This may indicate that the individual AZs lose more Ca²⁺channels as Cac is distributed among more AZs.

We have added a paragraph to the discussion dealing with these aspects. Lines 406-434.

The second question deals with the Brp signal. The authors show quite nicely that the Brp signal in the ORNs that is apposed to the PNs increases (Fig. 2U-V). The authors go on to state by electrophysiology that the ORN-to-iLN connection undergoes no such potentiation over development. But they do not report Brp levels in ORNs apposed to the iLN postsynapse.

If it were the case that those connections are not potentiated after cac knockdown, then one should expect no increases in Brp signal. However, if one did see an increase in Brp signal apposed to iLNs, then that would call into question the homeostatic mechanism proposed for the ORN-to-PN connection. In that scenario, the Brp signal would seem to be correlative, not causative.

The reviewer points out an important experiment. We have now imaged Brp-positive AZs at ORN-iLN synapses and have included this data in the new Supplemental Figure S11. The obtained results indicate that in fact two plasticity processes are induced by *cac*^{RNAi}:

- (i) At both ORN-PN and ORN-iLN synapses the fluorescence intensity of individual Brp punctae is increased (Figure 4, Supplemental Figure S11). As argued above this compensation likely prevents the release probability of individual AZs from dropping too far. Interestingly, the increased local Brp concentration observed here may well be related to the role of Brp in presynaptic homeostatic plasticity at the larval NMJ (see e.g. (Böhme et al., 2019; Goel et al., 2017; Mrestani et al., 2021; Weyhersmüller et al., 2011).
- (ii) The number of synapses only increases at ORN-PN connections. This is visualized by an increase in the number of Brp punctae (Figure 4). Moreover, nicotine puffing and mini amplitudes indicate that the additional presynaptic sites are matched by a corresponding increase in postsynaptic partners (Figure 4A-C). The new data show that this compensatory increase in synapse numbers does not occur at ORN-iLN connections (number of Brp punctae in Supplemental Figure S11). This result is consistent with the electrophysiology-based interpretation that *cac*^{RNAi} triggers a homeostatic increase in release site numbers specifically at ORN-PN connections. At present, we do not know what instructs the formation of additional ORN-PN synapses and in this context Brp may well be correlative. Addressing this aspect will certainly be of interest in future studies.

We have added these considerations to the discussion. Lines 409-429.

3. The variability after the knockdown of cac is interesting. To this reviewer, it seems to reflect a high variability that one sees in neurotransmission at low probabilities of release. In that sense, if one looks at the high odor concentration data (Fig. 1F) vs. the low odor concentration data (Fig. 1O) – just for wild type – one sees a much higher variance for the low concentration. So, it might not simply be the case that the system is broken at the high concentration and fixed at the low concentration. This reviewer wonders if an even higher odor concentration were presented in for 1F, would that ameliorate the difference observed? Alternatively, if a higher calcium concentration were used in the recording saline, would that be enough to erase the variance difference?

Side note: even though the average eEPSCs are not changed in Fig. 2B the cacRNAi data look more highly variable than control.

We have performed the experiments suggested by the reviewer. We first examined response reliability to pure odors (Figure for Reviewers 1 A), an order of magnitude above those in Figure 1F. Following the trend from low to high odor concentrations, the *cac*^{RNAi}-induced effect of reducing the correlation was even stronger when pure odors were used (Figure for Reviewers 1B). However, surprisingly, *cac*^{RNAi} no longer had an effect on the variance for pure odors (Figure for Reviewers 1C).

Recalling our finding that monosynaptic mechanisms influence the correlation values, whereas circuit effects that recruit the local neurons control the variability of PN activity (new Figure 5), we reasoned that perhaps the lack of an effect on variance arises from the activity of the local neurons. We showed that local neurons of *cac*^{RNAi} flies hardly respond to an odor concentration of 10⁻¹ (new Figure 5). We therefore examined the response of local neurons to pure odors. Indeed, the use of pure odors almost completely restored local neuron activity of *cac*^{RNAi} flies (Figure for Reviewers 1D), thus explaining why *cac* knockdown no longer had an effect on the variance.

We completely agree with the reviewer about this comment. However, we feel that it would be difficult to incorporate the pure odor data, as its interpretation requires an understanding of circuit effects and the local neuron activity, which are only dealt with at a later stage of the manuscript. We therefore think that this addition would cause more confusion than benefit.

Since the reviewer comments along with our response are available to the readers upon publication of the manuscript, we respectfully request that the reviewer will allow us to present this experiment only in this response letter.

As suggested by the reviewer, we also attempted to use a high extracellular Ca²⁺ concentration for the electrophysiological recordings. However, using the high Ca²⁺ concentration, which will affect all components of the *in vivo* circuitry, resulted in almost complete block of the odor responses (Figure for Reviewers 1E) and did not allow us to complete the experiment.

Figure for Reviewers 1: Pure odor analysis

A. Peristimulus time histogram (PSTH) of PN population responses to three odors examined as indicated (shaded areas represent SEM, odor pulse is labeled with a black bar). Knockdown of *cac* (green) in ORNs resulted in decreased PN odor responses. Odors were not diluted ($n=50$ flies). *Orco-GAL4* was used to drive the RNAi constructs and *GH146-QF* drove *QUAS-GFP*.

B. Temporal reliability analysis. Pairwise correlations for each odor-neuron combination were pooled across all odors. Driving *cac^{RNAi}* in ORNs reduces correlation values to a greater extent than in Figure 1D.

C. Firing-rate reliability analysis. The increased variability which was detected in *cac^{RNAi}* flies using an odor concentration of 5×10^{-2} (Figure 1F) was not observed using pure odors.

D. PSTHs of iLN responses to isopentyl acetate. When presented with the pure odor, *cac^{RNAi}* flies (blue) displayed similar responses to *wt* flies which were presented with an odor concentration of 5×10^{-2} (black and green are the same as in Figure 5C). Pure odor *cac^{RNAi}*, $n=24$.

E. PSTH of PN population responses to isopentyl acetate in 5mM extracellular Ca²⁺. The odor responses were significantly lower than in recordings with a physiological Ca²⁺ concentration (Figure 1C). $N=25$.

Minor Points

1. For genetic manipulations, *cac* (or *UAS-cacRNAi*) is more correctly italicized and lowercase. For *Cacophony* protein, uppercase *Cac* is correct. I know this is the kind of minor point that could end up being a time-consuming thing to fix because it would involve changing multiple pieces of text in every single figure. So, to be clear, this is a suggestion for an edit – not a required edit that changes anything about the clarity of the paper or figures.

We have changed the text and figures accordingly.

2. The experiments testing 0 days vs. 2-4 are very helpful in terms of defining a temporal component to the plasticity. Do the authors happen to know when exactly the ORN-specific *Gal4* turns on? That could indicate how long the animals were experiencing a loss of *cac* expression.

According to previous publications *Orco-GAL4* is already expressed at the larval stage (Fishilevich and Vosshall, 2005). We now also directly examined expression at the pupal stage by imaging a *UAS-GFP* reporter. The strong GFP signal demonstrates that *Orco-GAL4* is also active at the pupal stage (Figure for Reviewers 2).

Figure for Reviewers 2: *Orco-GAL4* is expressed in the pupal stage. 2-photon imaging of the antennal lobe (yellow). Average intensity projection of a P10-P12 pupa expressing GFP in ORNs.

3. Figure 2U: I do not think the authors mean square meters (m^2). Square microns (μm^2) maybe?

We thank the reviewer for picking this up. We have corrected this.

4. The authors note in the Introduction and Discussion how forms of homeostatic plasticity have been characterized at synapses like the NMJ. Of interest to this study, it is known from that NMJ work that Cacophony is needed to promote homeostatic potentiation of release properties. On one level, that would seem to suggest an interesting connection to the current work. However, what is happening here seems different. In the present study, it seems like the loss of *cac* is the initial challenge to this circuit, not a substrate for compensation. At the NMJ, *Cac* appears to be a substrate of homeostatic plasticity signaling machinery after a different type of challenge. Linking the two, *Brp* seems to act like a shared substrate of compensation in both systems.

We appreciate the reviewer's insightful comment. This interesting aspect is likely related to the increased local *Brp* concentration at AZs induced by *cac*^{RNAi} (Main Point 2, see above). In the context of this observation, we have included a discussion of the role of *Brp* as a substrate for homeostatic plasticity both at the NMJ and in the antennal lobe. Also, we have now drawn attention to the fact that *Cac* can act as both a trigger and an effector of plasticity. Lines 425-434.

Reviewer #2

In this study the authors tested the impact of modulating pre-synaptic machinery of the olfactory receptor neurons (ORNs) on the temporal and rate coding of the olfactory stimuli in fruitfly antennal lobe projection neurons (PN). Overall, they observed that modulating pre-synaptic machinery reduces the PN odor response amplitude and increase the temporal variability of spike times, and jitter. They also went further and dissect these effects and found out that while rate code effects are mainly through multi-synaptic ORN to LN to PN synapse, the temporal code effects are monosynaptic for the ORN to PN synapse. Authors event dive deeper and identified the synaptic mechanisms of how GABAergic and Glutamtergic LNs can contribute to response reliability. Finally they tested the effects of these manipulations on olfactory behaviors and showed that while there were no differences in long odor exposure assays, when the odor exposure periods are reduced, animals with perturbed ORN pre-synaptic machinery performed significantly worse.

I think this is a technically well executed study, and very well written and easy to read manuscript. The authors did so many experiments that I had in mind, and as I read the manuscript, I felt like many of my questions are answered. I also sensed that the completeness of this nice story is partly due to some pervious rounds of revisions, which dig further into the mechanisms of the observed effects. Hence I support the publication of this manuscript, and I have only minor comments below, which I hope that it will help authors to clarify few things.

1) In general, the entire study relies on population effects in unidentified PNs. While I understand that it will be very difficult to use the Gal4 system both for identifying PNS and manipulating ORNs, is there no pharmacology that could target this pre-synaptic machinery while identifying a single type of stereotyped PN, and conclusively show the effect on spike reliability in genetically identified PNs?

We agree with the reviewer that this experiment would be very interesting. However, to date the genetic tools required to complete the experiment are not available and thus this work cannot be performed in the time frame of the revision. In addition, pharmacological interventions will affect other circuit elements, which may occlude the conclusions from these experiments. We certainly agree that this is an important question that should be pursued in a dedicated follow-up manuscript.

2) Authors discuss a lot about trial to trial variability, and its effect in odor coding. A recent study in fruitfly PNs (<https://doi.org/10.1016/j.celrep.2021.110165>) have shown that repeated odor exposure increases reliability of odor representations and reduced ongoing spontaneous activity. It might be interesting to see whether the authors observe such increased reliability after repeated exposure depends on the AZ mechanism that is perturb in this paper.

We thank the reviewer for this nice idea. We have performed this experiment and indeed found both the increase in reliability in *wt* flies and that *cac*^{RNAi} abolishes this increase. This is presented in the new Supplementary Figure S9.

3) Authors uses correlations as a measure for similarity but not Euclidean distance, could this help to better interpret the amplitude effects on the odor responses?

We thank the reviewer for this comment. We have added an analysis with Euclidean distance and found that it provides similar results as the correlation analysis. This is now presented in the new Supplementary Figure S5.

4) Alternatively cosine distances can be a reliable measure for not amplitude dependent reliability across stimulus delivery. I think that the authors should test their reliability analysis by using multiple complementary measurements of reliability.

We have attempted to also use the cosine distance measurement. However, in *cac*^{RNAi} flies many of the odor responses were very low yielding vectors containing mostly zero values. These vectors are not suitable for the cosine measurement. Indeed, whereas for measurements from *wt* flies the cosine distance provided the expected results, in the case of *cac*^{RNAi} flies, strong noisy fluctuations were observed that arose from the presence of “zero vectors” (Figure for Reviewers 3).

Figure for Reviewers 3: Cosine distance analysis. Pairwise cosine distance for each odor-neuron combination were pooled across all odors for the data in Figure S1.

5) Spike reliability probably a lot more important for low odor concentrations, isn't it? Why the effect is more prominent as high concentrations? Does this have anything to the way the reliability is analyzed?

We agree with the reviewer that spike reliability would seem to be more important for low odor concentrations. However, the behavioral results with 0 day-old flies seem to point to the importance of spike reliability at all concentrations. In addition, this manuscript does not examine in which activity regime spike reliability is more important but rather it examines which conditions are more sensitive to the perturbations of the release machinery. We find that conditions that drive synaptic release to the limit (i.e. high odor concentrations) are, as expected, more sensitive to the perturbation and cannot be fully compensated by the homeostatic plasticity, contrary to more mild conditions (i.e. low odor concentrations). In addition, we find the same reduction in reliability for both the low and high odor concentration in 0-day old flies, before homeostatic compensation takes place. This indicates that the differences observed between low and high odor concentrations after homeostatic compensation has occurred, do not arise from the analysis method, but rather from a genuine biological process.

6) In general it will help a lot to give a little more information in the main text about the quantifications, since it is not easy to go back and forth between legends and the main text to understand what exactly the authors did.

We have now added more explanations to the text and refer to the methods where appropriate. Lines 114-134 and 573-589.

7) I also found some of the figures are not well arranged or labelled, some polishing of figures could make it easier to read this manuscript.

We have adjusted and split the figures.

8) Markus Meister lab have investigated encoding of visual information based on the timing of the first spike of retinal ganglion cells. The authors touch up on some of these concepts in this manuscript. I think it is certainly worth to cite these previous studies.

The reviewer is absolutely correct. We have added references to the relevant papers (Berry et al., 1997; Gollisch and Meister, 2008).

9) *Analysis in Figure 1E is not well referenced or explained in the text nor in the methods. Please try to clarify better.*

We have now added an explanation to the text and refer to the methods. Lines 114-134 and 573-589.

10) *Figure 1F, please plot the time bins in ms, not in number of bins.*

We thank the reviewer for this comment. We have corrected this in all cases where this type of graph is shown.

11) *Figure 1L-O, is there a significant increase in correlations in AZ perturbed animals? not discussed?*

Although there is a significant difference here, the effect size of this difference is very low (0.08) compared to Figure 1D, E (-0.21). In addition, this increase does not affect the flies' behavior (Figure 6E, F). We have therefore not discussed this difference.

12) *is there a significant effect in Figure 3K, I can not find any statistics*

The data in former Figure 3K is now presented in new Figure 5K. The actual probability curves for the correlation in Figure 5K, on which the statistical analysis is performed is presented in the new Supplemental Figure S12. We have added the statistics to Figure S12 and to Table S1.

Reviewer #3:

This study defines in vivo roles for presynaptic calcium channels, presynaptic release probability and homeostatic mechanisms in neural coding reliability. This is achieved through cell type-specific genetic manipulation and rigorous electrophysiological analysis of a well-defined neural circuit within the Drosophila CNS. The work is convincing and impressive. It builds on previous studies to establish an attractive model for in vivo genetic analysis of neural circuits in the context of behavior, and the results include several novel and noteworthy findings of interest to the broader neuroscience community.

Strengths:

1) Approach. Despite significant technical challenges, the authors combine cell type-specific genetic manipulation with patch clamp recording and imaging of identified cell-types within a defined neural circuit. Moreover, examining odor responses at the circuit and behavioral levels provides a compelling in vivo physiological context for this work.

2) Key findings. KD of cac resulted in a combination of functional and compensatory morphological changes within the olfactory circuit. This might have been viewed as confounding, but the authors dug in and addressed these changes in a systematic way. In the process, they made noteworthy discoveries, including a first in vivo look at how release probability and homeostatic mechanisms can influence neural coding.

cac KD produced changes in coding reliability within the olfactory circuit. After these were characterized in the context of two distinct components, variability and temporal reliability, the underlying mechanisms were examined at the synaptic and circuit levels. cac KD did not change EPSC amplitude, however the authors noted an increase in facilitation, which is suggestive of decreased release probability. Accordingly, they considered that normal EPSC amplitudes may result from a compensatory change in synapse number or quantal size. Analysis of mEPSCs and immunocytochemical studies confirmed the former possibility. This thoughtful and systematic approach demonstrated homeostatic plasticity within the circuit and presented an opportunity to examine its role in coding reliability. Further analysis examined the different cell and synapse types within the circuit and their contributions using genetic and pharmacological methods. Although these are complex and challenging problems, even in a simple circuit, the authors addressed them experimentally and made a solid case for their interpretations. In the end, this study provides a detailed dissection of how synaptic mechanisms, circuit interactions and homeostatic plasticity can influence the components of coding reliability in distinct ways.

3) Model. Another strength of this study is its further development of an attractive model for in vivo genetic analysis of homeostatic mechanisms in the CNS. Studies in this model can address homeostatic plasticity within a defined neural circuit and physiological context and, importantly, in a fully developed, mature organism with established synaptic and circuit architecture.

4) The manuscript is clear and well-organized. The authors' efforts to provide a conceptual

framework for understanding technical aspects of their work, including analysis of coding reliability, will be helpful to a general audience. The results are presented with a narrative that conveys the rationale for each step in their analysis. Finally, the authors emphasize complementary findings from different experimental approaches, and this further strengthens their conclusions.

We thank the reviewer for these kind words.

Weaknesses (none) and minor suggestion:

This careful study reports complexities in genetic analysis that are easily overlooked or neglected. Homeostatic compensation for cac KD, and the resulting changes in synapse and circuit morphology, led to important findings in this study and cannot be considered a weakness. However, it does preclude simple conclusions about the specific role of cac and presynaptic release probability in neural coding under normal conditions. This caveat might be acknowledged in the discussion. It would be of interest if future studies in this model can examine the acute physiological consequences of disrupting cac and presynaptic release probability. If feasible, this might be achieved by developing cell type-specific tools based on an existing temperature-sensitive cac mutant.

Yes, the reviewer raises a very good point. In our discussion we now draw attention to the fact that the homeostatic compensation prevents a simple interpretation of how *cac* and neurotransmitter release probability shape neural coding. Lines 389-393.

Indeed, it would be of great interest to test how an acute disruption of *cac* would affect the circuit, perhaps via a cell-type specific dominant negative effect of a transgenic construct. We agree that the temperature sensitive *cac* variant would be ideal for a follow-up study. We have now also added this consideration to the discussion.

Rick Ordway

References

- Böhme, M.A., McCarthy, A.W., Grasskamp, A.T., Beuschel, C.B., Goel, P., Jusyte, M., Laber, D., Huang, S., Rey, U., Petzoldt, A.G., et al. (2019). Rapid active zone remodeling consolidates presynaptic potentiation. *Nat. Commun.* 10.
- Fishilevich, E., and Vosshall, L.B. (2005). Genetic and functional subdivision of the *Drosophila* antennal lobe. *Curr. Biol.*
- Goel, P., Li, X., and Dickman, D. (2017). Disparate Postsynaptic Induction Mechanisms Ultimately Converge to Drive the Retrograde Enhancement of Presynaptic Efficacy. *Cell Rep.* 21, 2339–2347.
- Kittel, R.J., Wichmann, C., Rasse, T.M., Fouquet, W., Schmidt, M., Schmid, A., Wagh, D.A., Pawlu, C., Kellner, R.R., Willig, K.I., et al. (2006). Bruchpilot promotes active zone assembly, Ca²⁺ channel clustering, and vesicle release. *Science* (80-). 312, 1051–1054.
- Mrestani, A., Pauli, M., Kollmannsberger, P., Repp, F., Kittel, R.J., Eilers, J., Doose, S., Sauer, M., Sirén, A.L., Heckmann, M., et al. (2021). Active zone compaction correlates with presynaptic homeostatic potentiation. 37, 109770.
- Weyhersmüller, A., Hallermann, S., Wagner, N., and Eilers, J. (2011). Rapid active zone remodeling during synaptic plasticity. *J. Neurosci.* 31, 6041–6052.

REVIEWERS' COMMENTS

Reviewer #1 (Remarks to the Author):

The revision manuscript Rozenfeld et al. was highly responsive to the comments that this reviewer and other reviewers made from the first round. In response to the original comments, the reviewers have added new experiments, most importantly demonstrating that synapsin knock down phenocopies cac knock down in terms of the reliability of neuronal coding and behavior.

New experiments have also sharpened the authors' conclusions about active zone components Brp and Cac and how they respectively may act as a substrate or a trigger for this form of plasticity. Finally, this reviewer also appreciates the pure odor experiment that the authors conducted and agrees that the overall context may be out of the bounds of this particular study.

I apologize to the authors and the editors for the extra time I needed to read and review this revision. I do appreciate the care that the authors put into the original story and into the revision. And I will reiterate, I think the data will be interesting to a variety of neuroscientists, including those who study circuitry, those who study behavior, and those who study synaptic plasticity and neurotransmission.

Reviewer #2 (Remarks to the Author):

I read the authors rebuttal and the revised manuscript. The authors successfully responded to my comments as well as other reviewers comments. I especially liked the several additional experiments (Synapsin knock down/behaviour, calcium imaging, structure of ORN-PN & ORN-LN connections)/analysis (trial to trial variability, response reliability), which strengthen the authors arguments, and highlight the importance of their results from a more general context.

I support the publication of this study.

Reviewer #3 (Remarks to the Author):

The authors have quite effectively provided additional experiment evidence and appropriate revisions to address the reviewer comments. In its current form, this manuscript is well-suited for publication in Nature Communications.